# Regulatory T cells promote cancer immune-escape through integrin αvβ8-mediated TGF-β activation

Alexandra Lainé[1], Ossama Labiad[1], Hector Hernandez-Vargas [1], Sébastien This[2], Amélien Sanlaville [1], Sophie Léon[3], Stéphane Dalle[1,4], Dean Sheppard [5], Mark A. Travis[6,7,8], Helena Paidassi [2] & Julien C. Marie [1✉]

Presence of TGFβ in the tumor microenvironment is one of the most relevant cancer immune-escape mechanisms. TGFβ is secreted in an inactive form, and its activation within the tumor may depend on different cell types and mechanisms than its production. Here we show in mouse melanoma and breast cancer models that regulatory T (Treg) cells expressing the β8 chain of αvβ8 integrin (Itgβ8) are the main cell type in the tumors that activates TGFβ, produced by the cancer cells and stored in the tumor micro-environment. Itgβ8 ablation in Treg cells impairs TGFβ signalling in intra-tumoral T lymphocytes but not in the tumor draining lymph nodes. Successively, the effector function of tumor infiltrating CD8[+] T lymphocytes strengthens, leading to efficient control of tumor growth. In cancer patients, anti-Itgβ8 antibody treatment elicits similar improved cytotoxic T cell activation. Thus, this study reveals that Treg cells work in concert with cancer cells to produce bioactive-TGFβ and to create an immunosuppressive micro-environment.

[1] Tumor Escape Resistance and Immunity department, Cancer Research Center of Lyon INSERM U1052, CNRS UMR 5286, Centre Léon Bérard, Claude Bernard Université Lyon 1, 69373 Lyon, France. [2] CIRI, Centre International de Recherche en Infectiologie, Université de Lyon, INSERM U1111, Université Claude Bernard Lyon 1, CNRS UMR5308, ENS de Lyon, 69007 Lyon, France. [3] Plateforme Ex-Vivo, Département de Recherche Translationnelle et d'Innovation, Centre Léon Bérard, Lyon, France. [4] Department of Dermatology, Claude Bernard Université Lyon 1, Centre Hospitalier Lyon Sud, 69495 Pierre Bénite, France. [5] University of California San Francisco, San Francisco, CA, USA. [6] Lydia Becker Institute of Immunology and Inflammation, University of Manchester, Manchester, UK. [7] Wellcome Centre for Cell-Matrix Research, University of Manchester, Manchester, UK. [8] Faculty of Biology, Medicine and Health, Manchester Academic Health Sciences Centre, University of Manchester, Manchester, UK. ✉email: julien.marie@inserm.fr

The tenet of tumor immunotherapy is based on the ability of the immune system to survey for malignant transformation and be efficient at eliminating cancer cells. However, solid tumors can escape from the immune system by orchestrating a microenvironment that limits an efficient anti-tumor immune response.

In the tumor microenvironment (TME), transforming growth factor beta (TGF-β) is regarded as a key cytokine-promoting potent immunosuppression[1]. Among the three isoforms of TGF-β (TGF-β 1–3), TGF-β1 is prevalent within tumors[2,3]. This polypeptide cytokine, highly conserved in all mammals[4], impairs numerous functions of effector T lymphocytes and promotes both development and stability of CD4$^{pos}$ Foxp3$^{pos}$ regulatory T cells (Tregs)[5,6]. Subsequently, the selective targeting of TGF-β signaling in T lymphocytes leads to an efficient elimination of cancer cells by effector T lymphocytes[7] repressing their cytotoxic functions[8]. Hence, neutralization of TGF-β-immunoregulatory effects has been thought of as a promising anticancer therapy. However, major safety issues were raised, one of which being the risk of unleashing massive autoimmunity, given the key role of TGF-β signaling in the repression of T-lymphocytes activation[5,6,9,10].

Importantly, TGF-β is one of the few cytokines secreted in an inactive form. This small latency complex is composed of the mature cytokine encircled by the latency-associated peptide (LAP), which are noncovalently associated. LAP covers all the contact sites of the mature cytokine that must interact with TGF-β receptor complexes (TGFβRI and TGFβRII) to induce TGF-β signaling, including the phosphorylation of SMAD2/3[11]. Within solid tumors, the latent TGF-β complex can be secreted by several cell types, including cancer cells, and Tregs[1]. Nevertheless, unlike the TGF-β produced by Tregs, TGF-β secreted by cancer cells seems essential for the repression of the anti-tumor immune response[12,13]. As long as LAP maintains close contact with the mature cytokine, the secreted latent TGF-β can be stored in the TME, attached to the extracellular matrix, without any immune-regulatory functionality[11]. Hence, activation of the secreted TGF-β latent complex, which involves exposure of the receptor-binding domain of the mature cytokine, is therefore indispensable for TGF-β- mediated immune-regulatory functions in tumors. Thus, deciphering the mechanisms by which the activation of TGF-β present in the TME occurs is essential to our comprehension of solid tumors escape the immune system and will highlight potential new effective anticancer therapies that specifically target TGF-β activation within the TME and thus limiting autoimmune side effects associated to the privation of TGF-β activation. In vivo, the activation of TGF-β1 is largely dependent on integrins, including the αvβ8 integrin, whose expression is regulated by that of the β8 subunit (Itgβ8)[14,15].

In this work, we demonstrate that the expression of the integrin αvβ8 in Tregs is essential to efficiently activate TGF-β produced by cancer cells and promote tumor immune escape. In the absence of expression of the β8 integrin chain (Itgβ8) in Tregs, TGF-β signaling is impaired in tumor-infiltrating effector T cells and their cytotoxic functions are unleashed leading to the efficient control of tumor growth. In patient tumors, treatment with a neutralizing anti-Itgβ8 antibody, as well as single-cell gene-expression analysis on tumor-infiltrating T cells, confirmed the relevance of our findings in mice to human pathology. Overall, our results reveal an unexpected collaboration between cancer cells and Tregs to create an efficient TGF-β-mediated immunosuppressive TME, highlighting that the targeting of Itgβ8 might constitute efficient immunotherapy.

## Results

**Itgβ8 is mainly expressed in regulatory T cells in tumors.** In order to understand the mechanisms leading to the activation of the latent complex in the tumor, we first analyzed Itgβ8 cellular expression in the TME. To monitor Itgβ8 by flow cytometry, we took an unbiased approach by generating an *Itgb8-td-Tomato* reporter mice, in which we previously validated that tdtomato-positive cells expressed Itgβ8 protein in different cell types, including T lymphocytes[16]. Flow cytometry analysis of tumors (melanoma and breast cancer) revealed that among host cells composing the TME, Itgβ8$^{pos}$ cells were mainly (85–95%) CD45$^{pos}$ hematopoietic cells (Fig. 1a, b). T lymphocytes (CD3$^{pos}$), and particularly the CD4$^{pos}$ Foxp3$^{pos}$ (Treg) subset, composed the main portion of hematopoietic cells expressing Itgβ8, with approximately 80% of Itgβ8$^{pos}$ CD45$^{pos}$ cells being CD4$^{pos}$ Foxp3$^{pos}$ irrelevant of the tumor type (Fig. 1c–f). Moreover, within the Treg compartment, we found that about 40-45% of cells expressed Itgβ8 (Fig. 1g, h) and only Itgβ8$^{pos}$ Tregs were endowed with the capacity to efficiently activate TGF-β1 (Fig. 1i) whereas both Itgβ8$^{pos}$ Treg and Itgβ8$^{neg}$Treg populations expressed similar levels of this cytokine (Fig. 1j). Thus, this first set of data reveals that Tregs constitute a large part of the Itgβ8-expressing host cells within the TME.

**Itgβ8 expression in Tregs impairs anti-tumor response and promotes tumor growth.** Next, in order to assess whether Itgβ8 expression by Tregs confers their abilities to control the anti-tumor immune responses by providing a bioactive source of TGF-β, we first selectively ablated *Itgb8* in Tregs, using *Foxp3-Cre Itgb8$^{fl/fl}$* mice (Foxp3$^{ΔItgβ8}$). Importantly, in Foxp3$^{ΔItgβ8}$ mice, Tregs retain their numbers, localization, as well as their suppressive functions, including the ability to produce TGF-β1. Moreover, no autoimmunity signs, neither uncontrolled effector T-cell activation have been observed in Foxp3$^{ΔItgβ8}$ animals[17,18].

Strikingly, in contrast to their littermate controls (Foxp3$^{Ctrl}$), Foxp3$^{ΔItgβ8}$ mice showed a profound impairment of tumor growth irrelevant of the tumor type (Fig. 2a–f). Notably, we observed that 25–50% of the Foxp3$^{ΔItgβ8}$ animals exhibited a complete control of the tumor progression depending on the tumor type (Table 1). Thus, Itgβ8 expression in Tregs promoted tumor growth, implying that the Itgβ8$^{pos}$ Treg population could affect the anti-tumor function of the effector cells, including T cells and natural killer (NK) cells.

To confirm this scenario, we next analyzed the immune compartment of tumors and that of their draining lymph nodes (tdLN). Interestingly, the proportion of NK cells and T cells, including Tregs, were similar in both TME and tdLN between Foxp3$^{ΔItgβ8}$ mice and Foxp3$^{Ctrl}$ animals (Supplementary Fig. 1A). In line with this observation, the proliferative status of T cells and NK cells was similar between Foxp3$^{ΔItgβ8}$ mice control animals in both tdLN and TME (Supplementary Fig. 1B). Thus, we ruled out a specific role of the Itgβ8$^{pos}$ Tregs in controlling proliferation, recruiting of effector immune cells into the TME as well as T-cell priming in tdLN. However, the inhibition of tumor growth observed in Foxp3$^{ΔItgβ8}$ mice was completely lost when animals were depleted of their CD8$^{pos}$ T lymphocytes (Fig. 3a, b), revealing CD8 T cells as the main effector cells in the control of tumor growth in Foxp3$^{△Itgβ8}$ mice.

Thus, altogether these observations suggested that Itgβ8$^{pos}$ Tregs exert their pro-tumoral effects by impairing the anti-tumor functions of CD8$^{pos}$ T lymphocytes. In agreement with this assumption, we observed that CD8$^{pos}$ T lymphocytes of the TME of Foxp3$^{ΔItgβ8}$ mice exhibited higher cytotoxic functions based on the production of granzyme B cytotoxic granules (GzB) in association with the surface expression of CD107 (Lamp1) compared to Foxp3$^{Ctrl}$ animals (Fig. 3c). Production of IFN-γ was also exacerbated in tumor infiltrating in both CD4$^{pos}$ T cells and CD8$^{pos}$ T cells from Foxp3$^{ΔItgβ8}$ mice compared to Foxp3$^{Ctrl}$

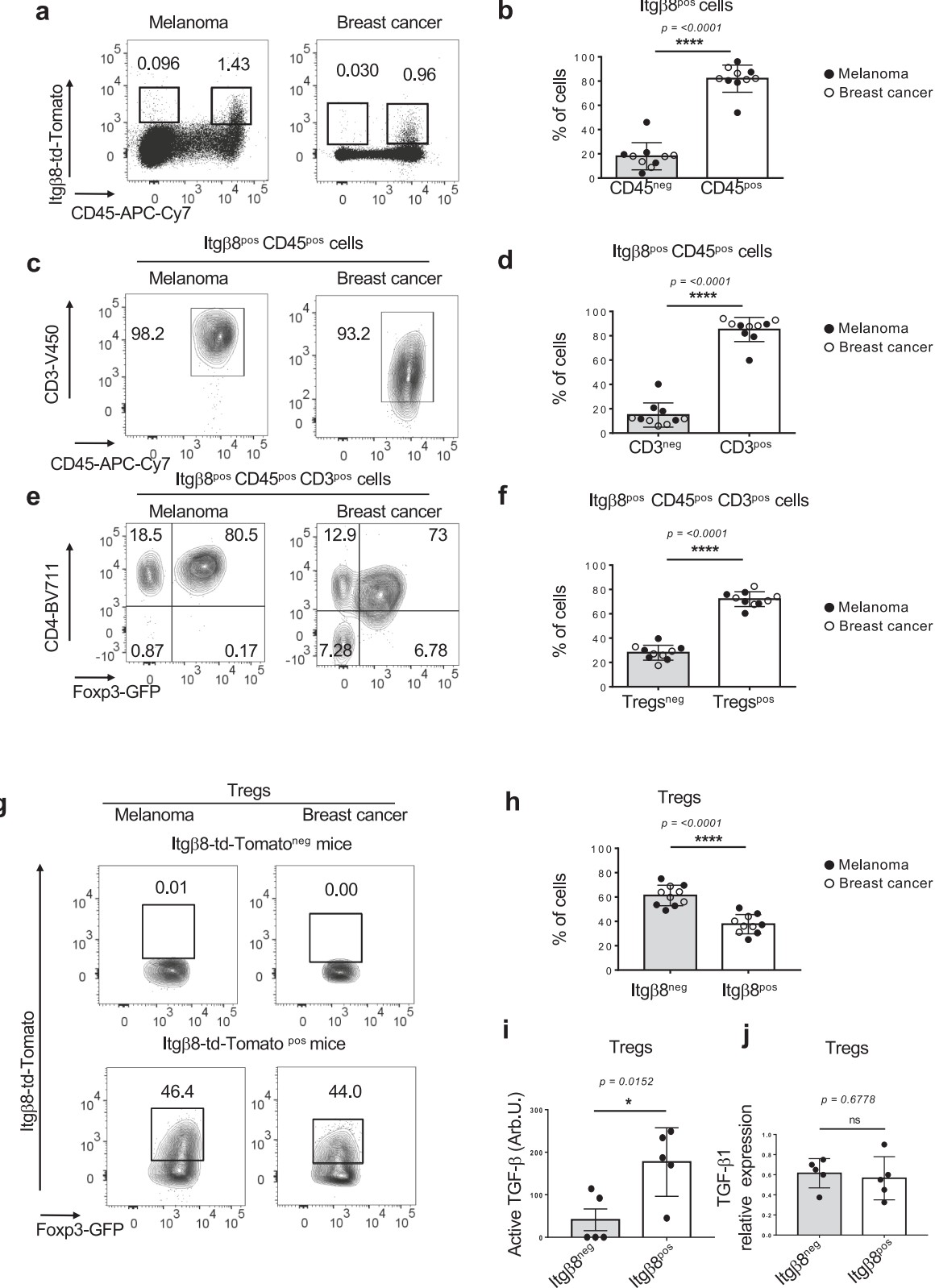

animals (Supplementary Fig. 2), implying that in addition of GzB production and the cytotoxic granule release, the exacerbated production of IFN-γ by T cells could also contribute to the tumor growth control in Foxp3$^{\Delta Itg\beta8}$ mice.

Supporting the exacerbated cytotoxic features of CD8$^{pos}$ T lymphocytes in the TME of Foxp3$^{\Delta Itg\beta8}$ mice, as well as the control of tumor growth in these animals, histology analysis showed higher numbers of apoptotic cells in tumors from Foxp3$^{\Delta Itg\beta8}$ mice than control animals (Fig. 3d, e). Importantly, in clear contrast to the TME, we failed to find any exacerbation of the cytotoxic phenotype of CD8$^{pos}$ T cells in the tdLN of Foxp3$^{\Delta Itg\beta8}$ mice (Fig. 3f). This observation, combined with the absence of systemic T-effector cell activation in secondary lymphoid organs of in Foxp3$^{\Delta Itg\beta8}$ mice[17,18], reveals a specific

**Fig. 1 Tregs compose the main cells expressing Itgβ8 in tumors.** *Itgb8-td-Tomato* reporter mice were injected with melanoma cells (B16) or breast cancer cells (E0771) in the dermis or in the mammary gland respectively. Eighteen days later tumors were analyzed by flow cytometry. Percentages of the gated populations are mentioned on dot plots and counterplots. **a** Representative plots illustrate the *Itgb8*-tdTomato expression in tumors. **b** Histograms demonstrating the percentages of Itgβ8pos cells among the hematopoietic compartment (CD45pos) and the nonhematopoietic compartment (CD45neg) in five tumors of each. **c, e** Representative contour plots illustrating the proportion of CD3pos cells among *Itgb8*-td-Tomatopos CD45pos cells and CD4pos Foxp3pos (Tregs) among *Itgb8*-td-Tomatopos T cells. **d–f** Histograms represent the average percentage of T cells among the *Itgb8*-tdTomatopos CD45pos cells and the percentage Tregs among the *Itgb8*-tdTomatopos T cells in five tumors of each. **g** Representative contour plots illustrating the *Itgb8*-tdTomato expression in Foxp3pos CD4pos T cells. **h** Histograms illustrate the average percentage of Tregs expressing *Itgb8*-tdTomato from five tumors of each. **i, j** *Itgb8*-tdTomatopos Tregs and *Itgb8*-tdTomatoneg Tregs were FACS-sorted, and their ability to activate TGF-β using TGF-β signaling reporter cells was measured (**i**), as well as their ability to express *Tgfb1* by RT-qPCR. Graphs illustrate the levels of bio-activated TGF-β expressed in arbitrary unit (Arb.U.) (**i**), as well as the levels of expression *Tgfb1*mRNA normalized on *Gadph*. Error bar, mean ± SD. **b, d, f, h** are representative of two experiments with five mice injected with each type of tumors per groups Data in (**i, j**) are representative of two experiments with five animals in total, each mouse providing both types of cells. *$P < 0.05$ paired two-tailed Student's *t* test. ns statistically not significant. ****$P < 0.0001$ were determined by unpaired two-tailed Student *t* test. Source data are provided as a Source Data file.

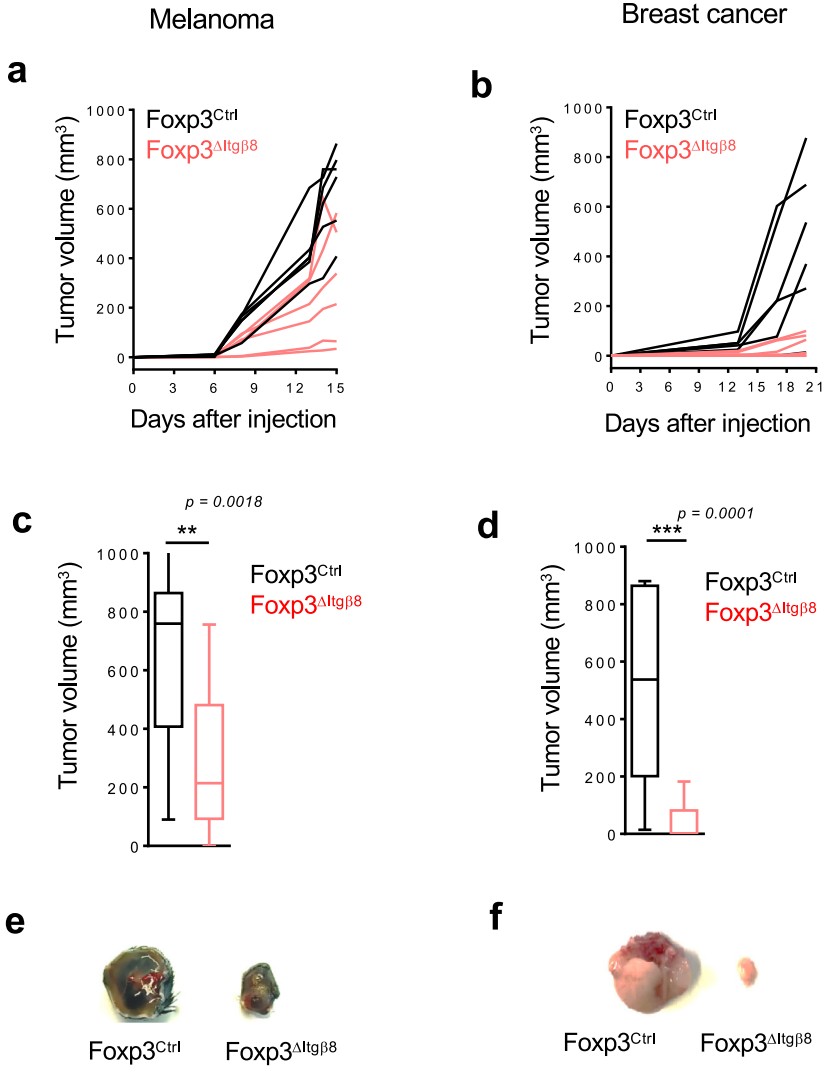

**Fig. 2 Itgβ8 expression on Tregs promotes tumor growth.** Foxp3ΔItgβ8 mice in red and their littermate controls in black (Foxp3Ctrl) were injected either i.d. with melanoma (B16 cells) or in the mammary gland with breast cancer (E0771 cells). **a, b** Representative graphs illustrating the size of the tumors at different days post-injection. $n = 5–8$ mice per group. Data representative of two independent experiments. **c, d** Graphs demonstrate the tumor size (mean ± SD), at the day of the euthanasia (18 days after injections) of the two independent experiments with a total of 16 mice per group. **c** $P = 0.0018$, **d** $P = 0.0001$. ***$P < 0.001$ and **$P < 0.01$ were determined by unpaired two-tailed Student *t* test. **e, f** Representative pictures of tumors at the end of the experiments. Source data are provided as a Source Data file.

**Table 1 Absence of Itgβ8 on Tregs represses tumor growth.**

| Genotype | Melanoma | | Breast cancer | |
|---|---|---|---|---|
| | Foxp3$^{Ctr}$ | Foxp3$^{\Delta Itg\beta8}$ | Foxp3$^{Ctr}$ | Foxp3$^{\Delta Itg\beta8}$ |
| Number of mice | 16 | 16 | 16 | 16 |
| Number of mice with non-measurable tumor | 0 | 4 | 1 | 8 |
| Percentage of mice with non-measurable tumor | (0%) | (25%) | (6.25%) | (50%) |

Foxp3$^{\Delta Itg\beta8}$ mice and their littermate controls (Foxp3$^{Ctrl}$) were injected either i.d. with melanoma (B16 cells) or in the mammary gland with breast cancer (E0771 cells). Table showing the percentage of Foxp3$^{\Delta Itg\beta8}$ mice and Foxp3$^{WT}$ mice with a completed control of tumor growth (tumor < 100 mm$^3$ all along the experiments) for both melanoma and breast cancer 18 days after implantation.

role for Itgβ8$^{pos}$ Tregs in the repression of the cytotoxic functions of CD8$^{pos}$ T lymphocytes selectively in the TME.

Altogether, these data identify Itgβ8 as a key mediator of Treg-induced suppression of the anti-tumor cytotoxic function of CD8$^{pos}$ T cells present in the TME with direct consequences on tumor progression.

**Itgβ8 expression on Tregs promotes TGF-β signaling controlling effector tumor T cells.** Given the role of αvβ8 in TGF-β activation[11], and the unique ability of the Itgβ8$^{pos}$ Treg subset to activate TGF-β1 compared to Itgβ8$^{neg}$ Tregs (Fig. 1i), we next assessed whether the repression of CD8$^{pos}$ T-cell cytotoxic functions in the TME of Foxp3$^{\Delta Itg\beta8}$ mice was due to an increase of the TGF-β signaling in the effector cells in the tumor. This assumption was even more motivated by the fact that, we found that the percentage of T cells with high activation of TGF-β signaling pathway, monitored by the phosphorylation of SMAD2-3, was halved in the TME of Foxp3$^{\Delta Itg\beta8}$ mice compared to Foxp3$^{Ctrl}$ animals (Fig. 4a). Notably, in contrast to the TME, and in line with the absence of T-cell overactivation in tdLN of Foxp3$^{\Delta Itg\beta8}$ mice the levels of phosphorylation of SMAD2-3 in T lymphocytes from tdLN were similar between Foxp3$^{\Delta Itg\beta8}$ mice and Foxp3$^{Ctrl}$ animals (Fig. 4b). Hence, Itgβ8$^{pos}$ Tregs are responsible for the increase of TGF-β signaling in T cells present in the TME.

In order to confirm that the exacerbated cytotoxic features of CD8$^{pos}$ T cells in the TME of Foxp3$^{\Delta Itg\beta8}$ mice were directly linked to the increase TGF-β signaling in effector T cells by Itgβ8$^{pos}$ Tregs, we developed genetic approaches allowing to sustain high levels of TGF-β signaling activation in effector T cells. The T-cell compartment of CD3ε deficient mice was reconstituted with purified Tregs from either Foxp3$^{Ctrl}$ mice or Foxp3$^{\Delta Itg\beta8}$ mice and Foxp3$^{neg}$ T cells expressing either a constitutively active (CA) form (TGFβRI$^{CA}$) or the unmodified form of TGFβRI (TGFβRI$^{WT}$)[19] (Fig. 4c). In TGFβRI$^{CA}$-expressing T cells, the TGF-β signaling pathway remains activated even in the absence of bioactive source of TGF-β in their microenvironment as we previously described it[19,20]. Similarly to the data illustrated in Fig. 3c, we observed that the absence of Itgβ8 expression in Tregs (Treg$^{\Delta Itg\beta8}$) increased the cytotoxic features of transferred wild-type CD8$^{pos}$ T cells. In contrast, in line with the ability of TGF-β to control GzB and CD107 expression[8], the maintenance of TGF-β signaling in effector T cells was sufficient to completely prevent the overactivation of their cytotoxic program as well as the repression of tumor growth we routinely observed in the absence of Itgβ8 expression on Tregs (Fig. 4d–g). Thus, within the TME, Itgβ8 expression on Tregs

increases the levels of TGF-β signaling activation in effector T lymphocytes which is sufficient to repress their cytotoxic functions.

**Activation of cancer cell-produced TGF-β1 by Itgβ8$^{pos}$ Tregs leads to tumor CD8 T-cell loss of function.** The aforementioned data, combined with the inability of Itgβ8 expression to modulate *Tgf-β1* expression in Tregs (Fig. 1i) and the minor role of TGF-β1 produced by Tregs in the control of the effector T-cell functions in the TME[12], strongly suggest that Itgβ8$^{pos}$ Tregs could contribute to the activation of TGF-β1 produced by other cells of the TME. As LAP reflects the inactive form of TGF-β, we evaluated the presence of LAP within the TME either in the presence or in the absence of Itgβ8 in Tregs. Strikingly, the classic fibrillar staining of the large latent complex was 2–3 times increased in the TME of Foxp3$^{\Delta Itg\beta8}$ mice compared to Foxp3$^{Ctrl}$ animals (Fig. 5a, b), without overexpression of TGF-b1 in TME of Foxp3$^{\Delta Itg\beta8}$ mice (Fig. 5c) revealing more inactive form of TGF-β1 were stored in the TME of these animals. In order to address, the source of inactive TGF-β1 which accumulate in the TME of Foxp3$^{\Delta Itg\beta8}$ mice, we selectively ablated *tgf-β1* in cancer cells regarded as high producer cells of TGF-β1 (TGF-β1$^{KO}$) in the TME[2,3] (Supplementary Fig. 3A). The accumulation of inactive form of TGF-β1 was lost in the TME of TGF-β1$^{KO}$ cancer cells (Fig. 5d, e). Of note, the absence of TGF-β1 production by cancer cells strongly impaired the tumor growth in wild-type mice but not in T-cell-deficient animals (CD3$^{KO}$) (Supplementary Fig. 3B, C). Confirming the importance of TGF-β1 produced by cancer cells in the control of T-cell anti-tumor immune response, the phenotypical analysis of CD8 T cells from the TME of TGF-β1$^{KO}$ cancer cells showed at 2–3 times exacerbation of their cytotoxic markers than in TME of TGF-β1 sufficient cancer cells (Fig. 5f, g). Importantly, the production of TGF-β1 by cancer cells had no significant impact Treg homeostasis (Supplementary Fig. 3D) and T-cell activation in the tdLN (Supplementary Fig. 3E, F). Thus, Itgβ8 expression by Tregs contributes to the activation TGF-β1 produced by cancer cells in the TME, with direct consequences on the repression the cytotoxic functions of CD8$^{pos}$ T cells present in the TME and thus on tumor immune escape.

**Itgβ8 expression on tumor-infiltrating T cells is associated with poor patient survival and CD8 T-cell activation.** We next analyzed the relevance of our data in mice to human pathology, particularly in melanoma patients. First, we confirmed that human T cells expressed *ITGβ8* in the TME by analyzing single-cell mRNAseq, and reported that *ITGB8* expression was prevalent in the Foxp3$^{pos}$ compartment of the TME of various tumor types, with 65-70% of Itgβ8$^{pos}$ T cells being Foxp3$^{pos}$ T cells (Supplementary Fig. 4). We then made use of publicly available sets of single-cell sequencing analysis data and obtained a specific gene-expression signature of Itgβ8$^{pos}$ T cells infiltrating the tumors, allowing us to perform multivariable survival analysis. We analyzed 358 patients bearing melanoma and revealed that high *ITGB8* score in tumor-infiltrating T cells was associated with poor survival (Fig. 6a). Of note, the poor survival prognostic associated with the presence of Itb8 Tregs was confirmed in other tumor types, except in colorectal cancer (Supplementary Fig. 5). The better survival prognostic observed in colorectal patients with high *ITGB8* score in Tregs from the TME was in agreement with the ability of Itgβ8$^{pos}$ Tregs to repress established chronic intestinal inflammation in mice[18] which was largely depicted to promote colorectal cancer progression[21]. Interestingly, our analysis confirmed that *FOXP3* expression alone in the T cells of TME was not sufficient to predict patient prognosis in any tumor types as previously showed[22] (Supplementary Fig. 5). Of note,

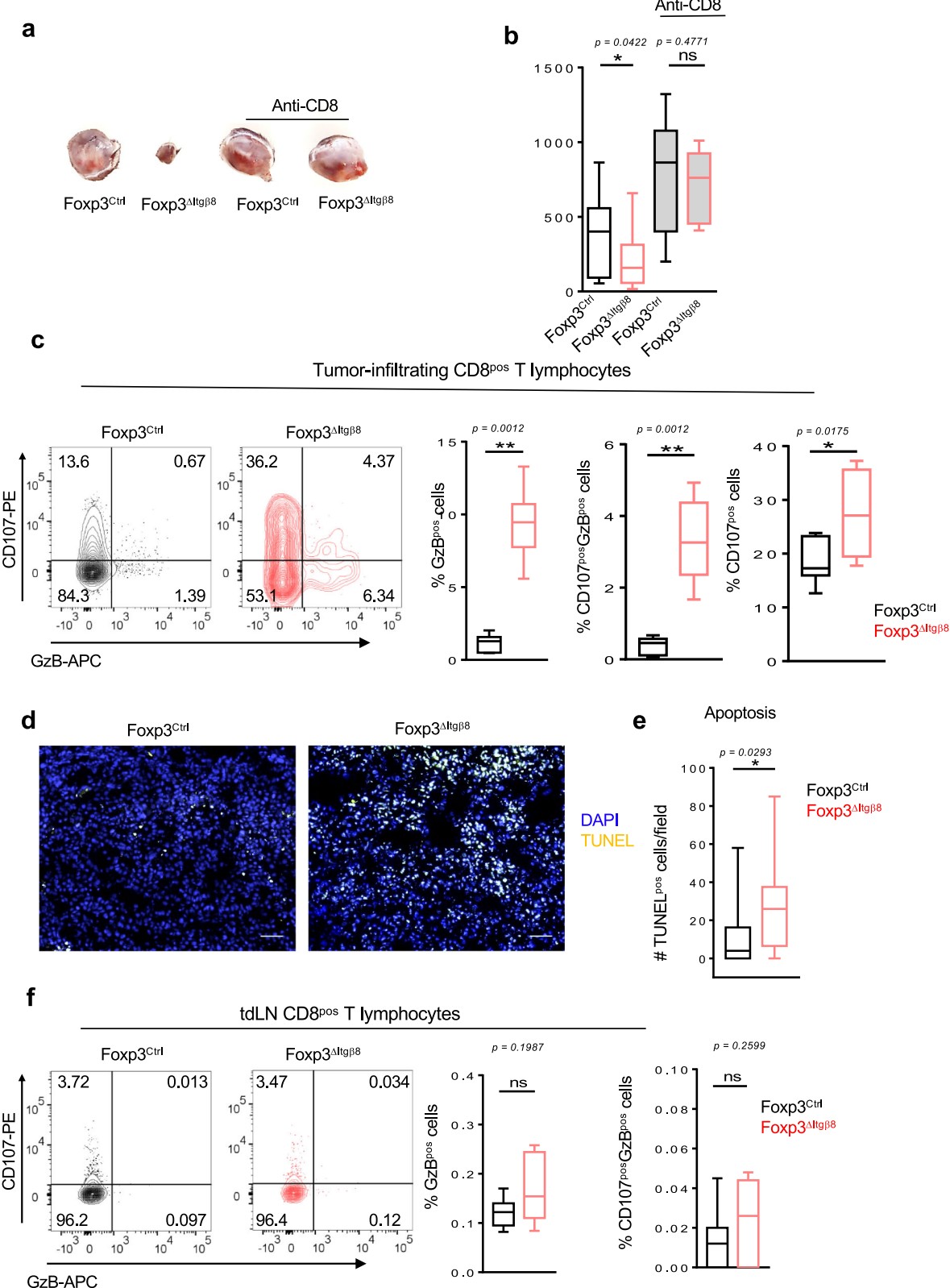

given that *ITGB8* expression was reported to be increased on activated human Tregs[18], we also removed the gene signature of activated Tregs in the *ITGB8* Treg signature and obtained similar survival prognostics as with the *ITGB8* Treg total gene signature for all the tumor types we analyzed (Supplementary Fig. 5). In line with poor survival associated with the presence of Itgβ8 Tregs in the TME of patients, we observed that the expression of *ITGB8*

Treg signature in the TME was inversely correlated with the activation of CD8 T cells present in the same TME (Fig. 6b). Thus, these data suggest that *ITGB8* expression in Tregs present in the TME might be useful as a predictor of poor patient survival and activation of CD8 T cells in tumors. Moreover, combined with our analysis in mice, the aforementioned observations suggest that neutralizing Itgβ8 ability to activate TGF-β in patient

**Fig. 3 Itgβ8 expression in Tregs impairs cytotoxic functions of tumor CD8$^{pos}$ T cells. a, b** Foxp3$^{Ctrl}$ mice, in black, and Foxp3$^{ΔItgβ8}$ animals, in red, were treated or not with depleting anti-CD8β to selectively remove CD8$^{pos}$ T cells. Animals were then injected i.d. with B16 cells. Pictures of tumors are representative of 6–11 animals from two independent experiments and the graph represents the tumor size at day 15 after tumor implantation. **c–f** Foxp3$^{ΔItgβ8}$ mice and their littermate controls (Foxp3$^{Ctrl}$) were injected either i.d. with B16 cells and tumors and their draining lymph node (dtLN) were analyzed by flow cytometry 15 days later (**c, f**). Counterplots illustrate the ability of CD8 cells to produce granzyme B (GzB) and degranulate based on CD107 cell surface expression CD8$^{pos}$ T cells infiltrating the tumor. Percentages of the gated populations are mentioned on counterplots. Histograms demonstrate the percentage of CD8$^{pos}$ T cells expressing GzB or co-expressing GzB and CD107 in different animals ($n = 7$) from two independent. **d, e** Representative pictures demonstrate TUNEL-positive cells (yellow nucleus), tumor slides were stained with DAPI (blue). The graph illustrates the density of apoptotic cells. Data are representative of two experiments, with 7–8 mice per group. Error bar, mean ± SD. For **b, c, f**, data are presented as mean ± SD. **b** $P = 0.0422$ (*)/$P = 0.4771$ (ns), **c** $P = 0.0012$ (**)/$P = 0.0175$ (*), **e** $P = 0.0293$. **$P < 0.01$, *$P < 0.05$. The paired two-tailed Student $t$ test for **b**, Unpaired two-tailed Student $t$ test for c, two-tailed Mann–Whitney for **c–f**. Scale bar: 50 μm. Source data are provided as a Source Data file.

tumors could be associated with stronger CD8 T-cells activation in the TME.

**Neutralization of Itgβ8 exacerbates cytotoxic T-cell function in TME of patients.** Finally, we assessed whether neutralizing Itgβ8 ability to activate TGF-β in patient tumors could affect effector T-cell's ability to respond to TGF-β and develop an efficient anti-tumor response in the TME. To this end, we used an ex vivo culture approach in which two serial sections of live tumors were cultured either in the presence or in the absence of neutralizing anti-Itgβ8 antibody (Fig. 7a). This technique allowed us to address the effects of the anti-Itgβ8 antibody on the same TME of the given same patient in which the immune system compartment and its interactions with the tumor tissues were conserved. After treatment, CD8$^{pos}$ T cells from the tumors were analyzed by flow cytometry (Fig. 7b). We first monitored the effects of the anti-Itgβ8 antibody treatment on TGF-β signaling in patient melanoma. In response to anti-Itgβ8 antibody, we observed a 30–50% of reduction in phosphorylation of SMAD2/3 in CD8$^{pos}$ T cells from TME demonstrating that neutralizing Itgβ8 in the human tumors affects the levels of TGF-β signaling in CD8$^{pos}$ T cells infiltrating the TME (Fig. 7c, d). Strikingly, we also observed a 2–5-fold increase of cytotoxic features of CD8$^{pos}$ T cells present in the TME in the majority of the melanoma after anti-Itgβ8 antibody treatment compared to untreated condition (Fig. 7e, f). Of note, similar observations were made in breast cancers in response to neutralizing anti-Itgβ8 antibody treatment (Supplementary Fig. 6). Thus, neutralizing Itgβ8 is sufficient to impair TGF-β signaling in CD8$^{pos}$ T lymphocytes infiltrating human tumors and boost their cytotoxic functions, opening the path toward clinical applications based on Itgβ8 targeting in cancer.

## Discussion

The presence of Tregs in the TME is usually associated with a weakness of the effector T-cell responses and poor prognosis in patients. Though Tregs do not need to produce their own TGF-β1 to repress the effector T-cell functions in the TME[12], this study reveals that Tregs, and particularly the Itgβ8$^{pos}$ population, are essential to increase the levels of activated TGF-β produced by cancer cells responsible for efficient repression of T-cell cytotoxic functions within the TME. Thus, this collaborative work between cancer cells and Itgβ8$^{pos}$ Tregs increases the ability of the cancer cells to escape the immune system and fosters cancer progression.

In certain cancers, *Itgβ8* expression is observed in tumor cells express and the forced expression of *Itgβ8* in cancer cell lines was associated with TGF-β1 activation in vitro as well as the impairment of metastasis growth and vascularization modifications of the tumors after their implantation in mice[23]. Our results do not exclude other cellular actors than Itgβ8$^{pos}$ Tregs participate in TGF-β activation in the TME. Indeed, we observed that

TGF-β signaling in T cells infiltrating the tumors is not fully abolished in Foxp3$^{ΔItgβ8}$ mice. However, no role of Itgβ8$^{pos}$ cancer cells has been assigned to the regulation of the CD8 T-cell cytotoxic functions in the TME[23]. Hence, we propose that once Itgβ8$^{pos}$ Tregs colonize the tumor, they help enforce the activation of latent TGF-β1 produced by cancer cells so far ensured by other cells of the TME, including cancer cells themselves in the tumors where they express αvβ8 integrin. This help from the Tregs allows the TME to reach the optimal activation of TGF-β1 which blocks the cytotoxic functions of CD8 T cells and promotes tumor immune escape. The control of TGF-β signaling in CD8 T cells by activating TGF-β1 is likely facilitated by the unique ability of Tregs to be in the close vicinity of CD8 T cells in the TME[24]. The ability of Itgβ8$^{pos}$ Tregs to activate TGF-β1 produced by cancer cells is in agreement with recent biochemical investigations on αvβ8-mediated TGF-β1 activation, made outside the Treg context, suggesting that the latent complex released by a given cell can be activated by αvβ8 integrin expressed by others[25]. Moreover, Tregs have been shown to be capable of acquiring at their surface latent complex produced by other cells[26].

Interestingly, the capacity of Itgβ8$^{pos}$ Tregs to increase the levels of bioactive TGF-β1, to ensure repression of the cytotoxic functions of T cells appears particularly of importance in the TME. Indeed, in clear contrast to the TME, the absence of Itgβ8 expression in Tregs failed to alter TGF-β signaling in effector T cells in the tdLN, implying that either other cells expressing Itgβ8 or other mechanisms, independent of the Itgβ8, play a key role in the activation of TGF-β in the secondary lymphoid organs. In line with this, while a modification on LAP of the RGD sequence recognized by αvβ8 integrin recapitulates the autoimmune syndromes observed in the absence of TGF-β1[14], no signs of autoimmunity nor immune disorders were described in Foxp3$^{ΔItgβ8}$ mice,[17,18]. Moreover, depending on the tissue, the predominant role of certain cells has been depicted in the activation of TGF-β. In the gut, Itgβ8$^{pos}$ dendritic cells appear as key activators of TGF-β, whereas in the skin this function seems to be more dependent on keratinocytes[27,28,29]. In addition, the inflammatory context favors the role of Itgβ8$^{pos}$ Tregs in activating latent TGF-β[18]. Whether some inflammatory factors present in the TME reduce the expression of Itgβ8 by other cells than Tregs or repress alternative mechanisms of TGF-β activation could be considered as suggested in the gut[30].

The secreted latent complex can be stored in the microenvironment of the secreting cells and thus be accessible to integrins[31]. Our data reveal that cancer cells as a major source of latent TGF-β complex stored in the TME which is activated by Itgβ8$^{pos}$ Tregs. However, we do not exclude that Itgβ8$^{pos}$ Tregs can also activate TGF-β once secreted. Indeed, one of the features of Tregs is to express at their surface high amounts of the protein GARP, which can bind latent complex, then present it to αvβ8 integrin and thus contribute to the activation of secreted latent TGF-β [26,32]. However, in contrast to the absence of *Itgb8*, the

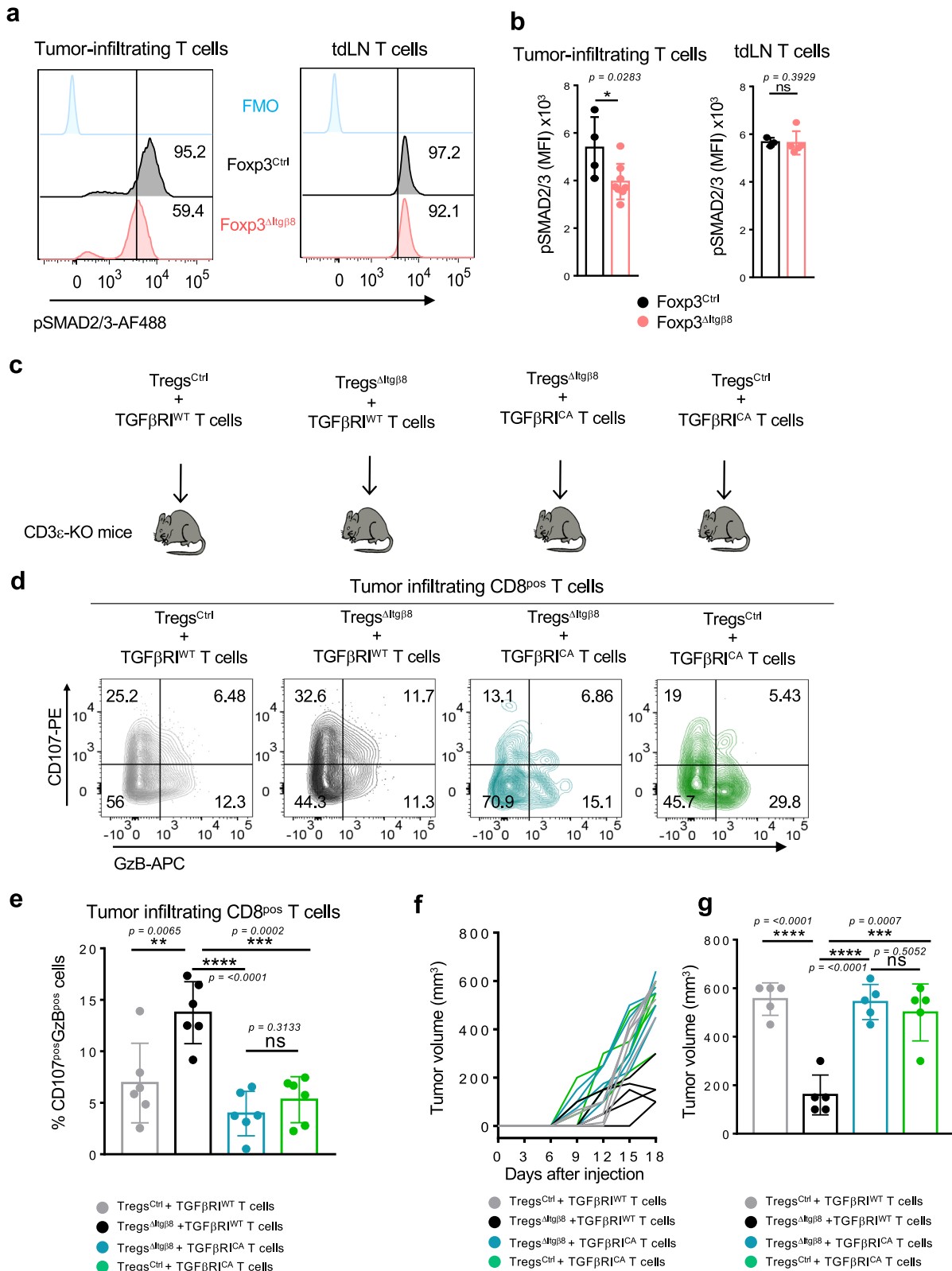

deletion in Tregs of *lrrc32*, which encodes for GARP, is not sufficient to affect tumor growth[33]. While GARP expression on Tregs contributes to the activation of secreted latent complexes, that of Itgβ8 contributes to the activation of latent complexes stored in the TME. Since much of the secreted latent complex of TGF-β is stored in the tissue[31], this could explain why the absence

of Itgβ8 expression in Tregs, and not that of GARP, is sufficient to influence TGF-β signal given to effector T cells and repress tumor growth.

TGF-β signaling is known to directly affect the CD8 T-cell cytotoxic function[8]. This study reveals that the activation of the latent complex by Itgβ8[pos] Tregs directly influences the levels of

**Fig. 4 Itgβ8$^{pos}$ Tregs promote TGF-β signaling in intra-tumor T cells affecting their anti-tumor response. a, b** Foxp3$^{Ctrl}$ mice (black) and Foxp3$^{ΔItgβ8}$ animals (red) were injected i.d. with B16 cells. Tumors and draining lymph nodes(tdLN) were harvested 15 days later and analyzed by flow cytometry. Representative histogram of the levels of phosphorylation of SMAD2/3 (p-SMAD2/3) in tumor-infiltrating T cells (**a**) percentage of cells with high phosphorylation of SMAD2/3 are indicated on each histogram as well as FMO control. Graphs demonstrate the mean of fluorescence intensity (MFI) ± SD in T cells (**a**). Data are representative of two independent experiments with 4–6 mice per group. **c–g** Purified Tregs from either Foxp3$^{Ctrl}$ mice or Foxp3$^{ΔItgβ8}$ mice were adoptively transferred with Foxp3$^{neg}$ T cells (CD3$^{pos}$GFP$^{neg}$) from either *CD4-Cre;Stop$^{fl/fl}$tgfbr1$^{CA}$,Foxp3$^{GFP}$* mice (TGFβRI$^{CA}$) or their littermate controls (TGFβRI$^{WT}$) in CD3ε$^{KO}$ mice. Ten days later, recipient mice were injected i.d. with B16 cells and their tumors were analyzed 16 days after their implantation by flow cytometry. **d** Representative contour plots of CD107 cell surface expression and GzB production by tumor-infiltrating CD8$^{pos}$ T cells are illustrated and percentages of each population mentioned populations are on the counterplots. **e** Graphs illustrate the percentage of GzB$^{pos}$ CD107$^{pos}$ cells and of CD107$^{pos}$ cells among CD8$^{pos}$ T cells infiltrating the tumors. Six mice of each group from two independent experiments were analyzed. Data are presented as mean ± SD. **f** Representative graph illustrating the size of the tumors at different days tumor cell post-injection. *n* = 5 mice per group. **g** Graph represents the tumor size at the end of the experiments. Data are representative of three independent experiments with five recipient mice per group. Means are shown ± SD. **b** *P* = 0.0283 (*)/*P* = 0.3929 (ns), **e** *P* = 0.0065 (**)/*P* = 0.3133 (ns)/*P* = <0.0001 (****)/*P* = 0.0002 (***), **g** *P* = <0.0001 (****)/*P* = 0.5052 (ns)/*P* = <0.0001 (****)/*P* = 0.0007 (***). **P* < 0.05, ***P* < 0.01, ****P* < 0.001 was determined by two-tailed Mann–Whitney test. ns statistically not significant.

TGF-β signaling delivered to intra-tumor effector CD8 T cells and thus their cytotoxic function. The restoration of TGF-β signaling in effector T cells fully prevents their cytotoxic functions associated with the deletion of Itgβ8 on Tregs and it confirms that the modulation of TGF-β signaling in effector cells by Itgβ8$^{pos}$ Tregs as the main mechanisms of action this regulatory subset in the TME. Based on several observations made in vitro and in the gut, TGF-β has been proposed to promote the conversion Foxp3$^{neg}$ T cells toward Foxp3$^{pos}$ cells in the tumor[34]. Interestingly, in the absence of Itgβ8$^{pos}$Tregs, the proportion and the numbers of Tregs remained unchanged within the TME. Moreover, the lack of *Tgfβ1* in cancer cells, which leads to the activate the cytotoxic function of CD8 T cells in the TME, failed to affect Treg homeostasis in the tumor. Hence, further investigations should confirm the ability of the activated TGF-β1 present in the TME to influence T effector cell conversion into Tregs. We recently reported that Itgβ8 expression on Tregs contributes to CD8 T resident memory cell (Trm) development which requires a bioactive source of TGF-β1[35]. Further investigations, using mouse models with slower growth than B16-F10 should address, whether in the context of TME, this Itgβ8 dependent function of Treg on Trm occurs and could contribute to long-term anti-tumor protection.

Targeting TGF-β effects on the immune cells in the TME is an important field of investigation for numerous companies. Our data strongly suggest that targeting Itgβ8 in patient could lead to potent activation of the T-cell cytotoxic program in the TME and control of the tumor progression. This idea is comforted by our ex vivo experiments, revealing that anti-Itgβ8 antibody treatment impairs TGF-β signaling in effector T cells and is sufficient to boost their cytotoxic functions in the TME of patients. Though the best way to efficiently target Itgβ8 effects in patients needs to define, the exacerbation of the cytotoxic functions of T lymphocytes selectively in the TME suggests that targeting Itgβ8$^{pos}$ Tregs may represent promising immunotherapy avoiding the risk of unleashing massive autoimmunity following a systemic neutralization of TGF-β effects[10].

In sum, this study reveals an unsuspected collaborative mechanism between cancer cells and Tregs with direct consequences on the repression of the anti-tumor function of effector T cells in tumors. Moreover, it provides evidence that targeting Itgβ8 could constitute a promising future anticancer immunotherapy in patients.

## Methods
**Mice**. *Itgb8-td-Tomato* mice were generated as described[16]. Generated animals were cross on *FOXP3-IRES-GFP* background[36] to follow Tregs. *FOXP3-Cre$^{eYFP}$;Itgβ8$^{-/-}$* (Foxp3$^{Ctrl}$) mice *FOXP3-Cre$^{eYFP}$;Itgβ8$^{fl/fl}$* (Foxp3$^{ΔItgβ8}$) mice[18],

*CD4-Cre;Stop$^{fl/fl}$;tgfbr1$^{CA}$;Foxp3$^{GFP}$* mice[19] were used. C57BL/6J mice (stock # 0632) and C57BL/6 CD3ε$^{KO}$ (CD3$^{KO}$) mice (stock # 020456) were purchased (Charles Rivers, France). Importantly, though Itgβ8 is mainly expressed in Tregs, we validated any leakiness of *Foxp3-CRE* construct in Foxp3$^{ΔItgβ8}$ mice, by breeding animals on Rosa26 reported background[37]. Animals were euthanized with euthanasia automate (Tem Sega) using 50–100% CO$_2$. All animals were between 2 and 6 months of age, all on a C57BL/6 background. Except for the breast cancer model, both genders were used without any differences between males and females. Foxp3$^{Ctrl}$ mice and Foxp3$^{ΔItgβ8}$ mice littermate were caged together with respect to gender. Mice were maintained in AniCan SPF mouse facility in Lyon, France, in stable temperature between 19–23 °C and 60–65% of hygrometry under a 12-h dark light cycle.

**Patient tumors and anti-Itgβ8 antibody treatment**. Primary breast adenocarcinoma tumors and primary melanoma were obtained by the Biological Resource Center of Centre Léon Bérard and Hospital Lyon Sud respectively. Primary melanoma, at noninvasive stages, was obtained after surgery in different regions of the body. Primary breast tumors, irrelevant of their hormonal status, were analyzed. No gender (melanoma) and age (breast cancer and melanoma) selection was performed to establish the patient cohort. Importantly, patients never received anticancer treatments prior to surgery. Fresh tumors were treated by the Ex-vivo facility of the Centre Léon Bérard Lyon France. They were embedded in the Ex-vivo facility-specific matrix gel© and cut at 250 μm with microtome (Seica). Tumor slides were then cultured on Uvac 1264 in RPMI-completed medium, 1% FCS, 1% HEPES, 1% penicillin/streptomycin, 1% MEM-NEAA (LifeTechnologies), 1% NaPyruvate (LifeTechnologies) with 20 μg/ml of neutralizing Itgβ8 antibody ADWA-16[38]. Tumor slices were harvested 48 h later, minced with scalpel, and incubated with 5 mg/ml collagenase IV (Gibco) and 1 mg/ml DNase I (Sigma, 11284932001) in RPMI supplemented with 1% FCS and 1% HEPES for 30 min at 37 °C with agitation prior cytometer analysis.

**Ethics**. Experiments on mice were performed in accordance with the animal care guidelines of the European Union, ARRIVE guideline, and French laws and were validated by the local Animal Ethics Evaluation Committee (CECCAPP) and the French ministry of Research (#9239 and #19584). Patient tumors were obtained after approval of the protocol by the institutional review board and ethics committee, with fully informed patient consent (French Ministry of Research agreement number: AC-2013-1871 and AC-2019-3426).

**Cell lines**. Breast medullary adenocarcinoma cell lines E0771(CVCL_GR23) and melanoma B16-F10 (B16) (CRL_6475) were obtained from ATCC. B16-shTGF-β1 cells were generated by the introduction of shTGF-β1 RNA in B16. Briefly, lentiviral vectors encoding shRNA pLKO.1 puro *Tgf-β1*: NM-011577.1-1753s1c1 and empty vectors were kindly given by Prof. D. Klatzmann (Paris). Infected B16 were selected on puromycin (Sigma, P8833). MLEC cells[39], with luciferase activity reporting TGF-β signaling, were used. All cell lines were maintained in DMEM (Gibco, 31966-021) supplemented with 10% FCS (LifeTechnologies, 10270-106), 1% HEPES (LifeTechnologies, 15630-056), 1% penicillin/streptomycin (LifeTechnologies, 15140-122). In total, 250 μg/ml of geneticin (LifeTechnologies, 10131-027) or 5 μg/ml of puromycin (P8833-10MG) were added for MLEC and B16-shTGF-β1 cell culture, respectively. All cell lines were tested negative for mouse pathogens, including mycoplasma by PRIA test (Charles Rivers).

**Measure of active TGF-β**. MLEC cells[39] were incubated with Itgβ8$^{neg}$ or Itgβ8$^{pos}$ Tregs. Luciferase activity was detected via the Luciferase Assay System (Promega, E1500) on a TECAN. TGF-β bioactivity is presented in arbitrary units (Arb.U.) after the withdrawal of the blank value of the medium alone.

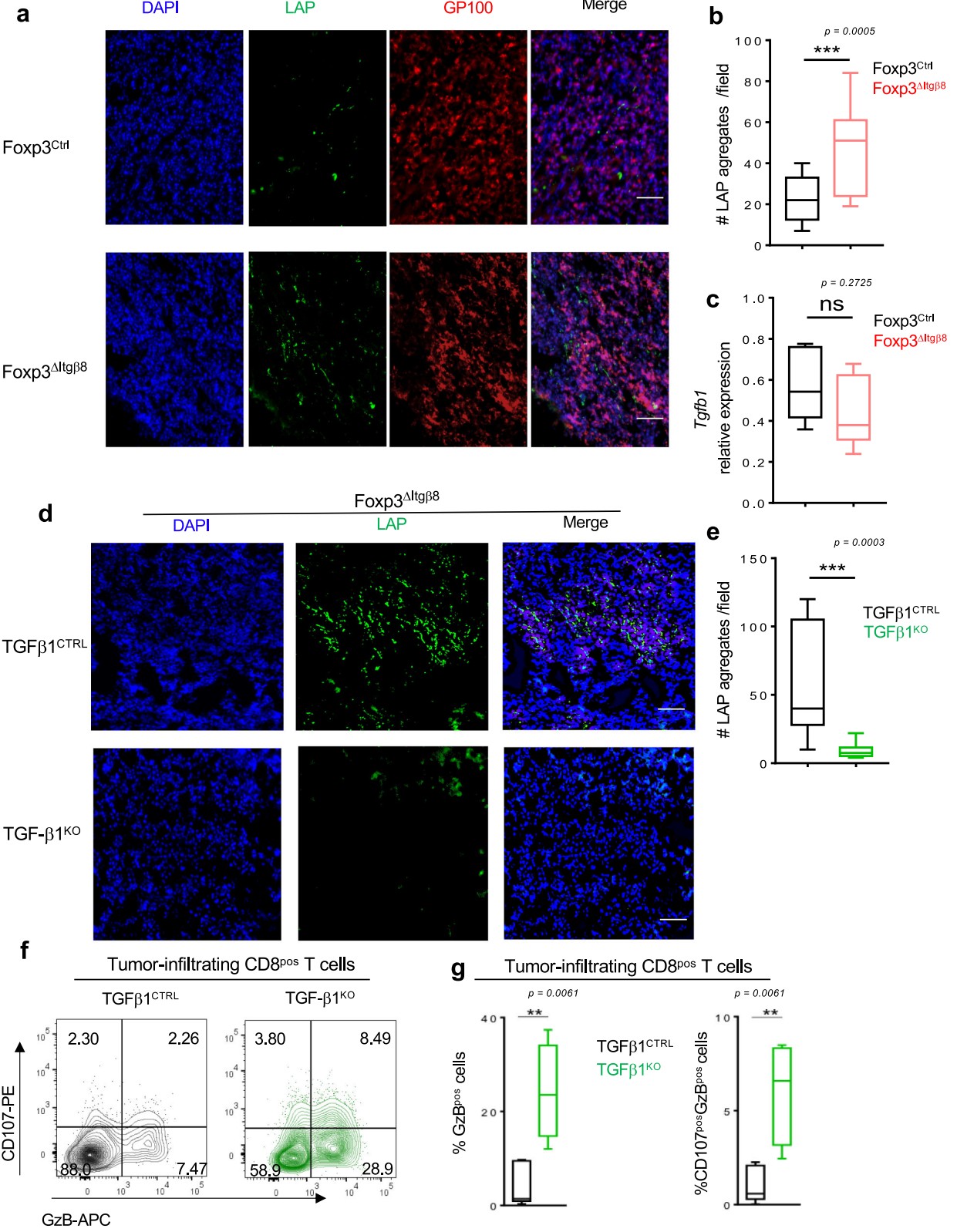

**Mouse tumor implantation and tissue preparation**. In total, $5 \times 10^5$ B16 cells were injected intra-dermally (id) in the back skin. In all, $5 \times 10^5$ E0771 cells were injected into the abdominal mammary gland # IV. Tumor growth was monitored every 3 days with a caliper in a double-bind manner. Tumor size (mm$^3$) was calculated as width × length × width. Tumors were minced with scalpel, and digested with 1 mg/ml collagenase IV (Sigma, C2674-1G) and DNAse I at 1 mg/ml (Sigma, 11284932001) in DMEM supplemented with 1% FCS and 1% HEPES. Tumor draining lymph nodes (tdLN, inguinal) were mechanically ground with glass slides.

**Adoptive T-cell transfers**. Foxp3$^{neg}$CD3$^{pos}$ cells from the lymph nodes of *CD4-Cre;Stop$^{fl/fl}$;tgfbr1$^{CA}$;Foxp3$^{GFP}$*mice, *CD4-Cre;Foxp3$^{GFP}$*mice and CD4$^{pos}$Foxp3$^{pos}$ cells from Foxp3$^{Ctrl}$ and Foxp3$^{\Delta Itg\beta 8}$ were purified by cell sorting with ARIA II (BD). In total, $5 \times 10^4$ CD4$^{pos}$Foxp3$^{pos}$ cells mixed with $4.5 \times 10^5$ CD3$^{pos}$Foxp3$^{neg}$ were intravenously injected to CD3ε$^{KO}$ mice.

**CD8$^{pos}$ T-cell depletion**. Depletion of CD8$^{pos}$ T cells was performed by intra-peritoneal injection of anti-CD8β (BioXCell, clone Lyt3.2; BE0223). In all, 150 μg

**Fig. 5 Itgβ8^pos Tregs activate TGF-β1 produced by cancer cells to repress tumor-infiltrating CD8 T cells. a–c** Foxp3^Ctrl mice and Foxp3^ΔItgβ8 animals were injected i.d. with B16 cells. 15 days later tumors were harvested and analyzed by immunostaining for LAP and melanoma (GP100), 11–13 field were analyzed for each tumor. The graph illustrates the numbers of LAP aggregates observed per field. **c** Histogram illustrates the expression of *Tgfb1* by RT-qPCR on isolated tumor cells after normalization with *gadph* expression. For **b**, **c**, data are presented as mean ± SD. **d**, **e** Foxp3^ΔItgβ8 animals were injected i.d. with either B16 *shTgfb1* (TGFβ1^KO) or B16 cells having received empty vector control (TGFβ1^CTRL), and tumors were harvested 15 days later and analyzed by immunostaining for the presence of LAP aggregates. Graph illustrates the numbers of LAP aggregates observed per field, with 11–12 fields were analyzed for each tumor. **f**, **g** Foxp3^Ctrl mice were injected i.d. with either TGFβ1^CTRL B16 cells (black), 4 tumors, or TGFβ1^KO B16 cells (green), 8 tumors, and CD8^pos T cells infiltrating the tumors analyzed 15 days later flow cytometry. Representative counterplots illustrating the cytotoxic functions are shown as well as graphs demonstrating the percentage of CD8^pos T cells expressing either granzyme B (GzB) or and GzB and CD107. **b** $P = 0.0005$ (***), **f** $P = 0.0061$ (**). All experiments were conducted on four mice per group in three independent experiments. Means are shown ± SD. ***$P < 0.001$ was determined by unpaired two-tailed Student $t$ test (**b**) and means are shown ± SD. **$P < 0.01$ was determined by a two-tailed Mann–Whitney test (**c**, **e**). ns statistically not significant. Scale bar (50 μm). Source data are provided as a Source Data file.

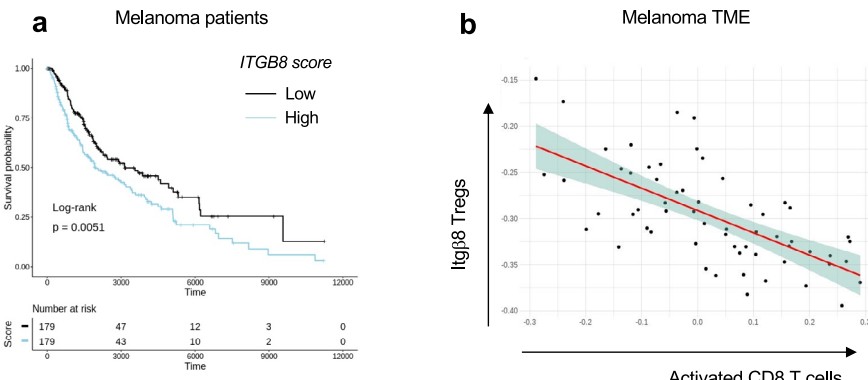

**Fig. 6 High Itgβ8 score on T-cell infiltrating melanoma correlates with patients' worse prognostic and CD8 T-cell activation in the TME.** Transcriptome signatures of Itgβ8^pos Treg cells from patient melanoma were extracted from single-cell RNA-seq data (as described in "Methods"). The extracted signatures were used for testing their association with overall survival on The Cancer Genome Atlas (TCGA) RNA-seq data from melanoma. Tumor samples were classified into low and high, based on the expression on *ITGB8* score. **a** Graph illustrates the overall survival of cancer patients stratified by *ITGβ8* score in tumor-infiltrating T cells with time expressed in days. $n = 358$ patients, 179 patients with high expression, 179 patients with low expression log-rank $P = 0.0051$ using Mantel Cox test. **b** Graph illustrating the inverse correlation between with ITGβ8 Tregs core and the activated CD8 T-cell score in the same melanoma. Each dot represents one tumor. Spearman $R$ value of correlation: −0.67.

per mouse were injected 4 days prior to tumor injection and every 4 days all along the experiment. Depletion was systematically checked by flow cytometry on blood samples before tumor injection and on lymph nodes and on the tumors after the experiments using a different anti-CD8β clone (YTS156.7.7, Biolegend).

**Cell sorting and flow cytometry analysis**. Surface staining of mouse cells was performed using the following fluorescent-conjugated antibodies: CD3ε 1/200 (145-2C11; BD biosciences; 561389), CD3ε 2C11; BD biosciences 100210), CD3ε 1/200 (145-2C11; BD; 564378), CD4 1/200 (RM4-5; Biolegend 100540; eBiosciences; 56-0042-82), CD4 1/200 (GK1.5; BD; 563050), CD8α 1/200 (53.6.7; BioLegend; 100748), CD8α 1/200 (53.6.7; BD; 563234), CD8α 1/200 (53.6.7; BD; 563068), CD45 1/200 (30-F11; BD; 557659; BioLegend103116), CD107a 1/100 (eBio1D4B, eBiosciences, 13-1071-82), NK1.1 1/100 (PK136; Biolegend; 108748). For intracellular staining, cells were fixed and permeabilized using Fixation and Permeabilization Buffer kit (00-5523-00, eBiosciences) according to the manufacturer's protocol. Granzyme B 1/100 (GRB05 Invitrogen), Ki67 1/400 (SolA15; Thermo-Fisher; 46-5698-82) were used. For cytokine staining cells were incubated with brefeldin A (eBioscience), for four hours, IFN-γ 1/100 (XMG1.2 BD bioscience; 563376) staining was performed with Buffer kit (00-5523-00, eBiosciences). For p-SMAD2/3 staining, cells were immediately fixed with Fixation and Permeabilization Buffer kit (00-5523-00 eBiosciences) prior to staining and anti-p-SMAD2/3 1/100 (D27F4, Cell Signaling) was detected with goat anti-rabbit A488 1/100 (Life-Technologies, A11034). For cell sorting, T cells were enriched with Pan T cell isolation kit II mouse (Miltenyi Biotec) and then stained with CD4 (GK1.5; ThermoFisher) and CD8 (53.6.7; BD biosciences). Itgβ8-td-Tomato^Pos Foxp3^GFPCD4^pos cells, Itgβ8-td-Tomato^neg Foxp3^GFPCD4^pos cells, CD4^posFoxp3-^YFPpos cells and CD3^posFoxp3^YFPneg cells were sorted on FACS ARIA II. Human cell stainings were performed using the following fluorescent-conjugated antibodies: CD3 1/200 (UCHT1; BD biosciences; 562280), CD45 1/100 (HI30; BD Biosciences; 560566), CD4 1/20 (RPA-T4; LifeTechnologies; 47-0049-42), CD8 1/100 (SK1; BD biosciences; 565289), CD107a 1/50 (H4A3; BD biosciences; 561345) and Granzyme B 1/100 (GRB05 Invitrogen) p-SMAD2/3 (D27F4, Cell Signaling) was detected with goat anti-rabbit A488 1/100 (LifeTechnologies, A11034). All

samples were acquired on BD Fortessa and data were analyzed with FlowJo Software version X.

**Mouse tumor TUNEL and immune-fluorescence staining**. Tumors were embedded in Tissue tek OCT compound (Sakura Finetek) and snap-frozen; 10-μm-thick sections were cut with CryoStar NX50 (Microm Microtech France). For TUNEL staining, sections were permeabilized with 0.2% Triton (Sigma, T9284) and digested with Proteinase K (ThermoFisher, K182001). For positive control, sections were incubated with DNAse I at 1 mg/ml (Sigma, 11284932001). Sections were then incubated with biotin-16-dUTP (Sigma, 11093070910) and TUNEL enzyme (Sigma, 11767305001) in deoxynucleotidyltransferase buffer (1 mM CoCl₂ (Sigma, 15862-1ml-F), Tris-HCl, 200 mM sodium cacodylate (Sigma, C0250-25G), 0.125% BSA (Sigma, A7906-500G)) at 37 °C for 60 min. Sections were washed in stop buffer (300 mM NaCl (Sigma, S3014-1KG), 30 mM Sodium Citrate (Sigma, 71406-500 G)) and blocked with 2% BSA. Sections were then labeled with streptavidin–phycoerythrin (eBiosciences, 12-4317-87), CD8-A488 (53-6.7, BD biosciences). For immunostaining, tumor sections were fixed in 4% PFA (Sigma) and stained with rabbit anti-mouse GP100 1/100 (ab137078 Abcam) was detected with goat anti-rabbit A647 1/100 (LifeTechnologies; A31573) and or LAP-PE 1/100 (TW7-16B4, Biolegend, 141404). All sections were stained with DAPI (Euromedex, 1050-A) and mounted with Fluoromount (Sigma, F4680-25ml). All samples were acquired on Upright microscope Zeiss Axioimager (SIP 60549) and data were analyzed with Zen 2 (blue edition).

**Quantitative real-time PCR**. mRNAs were isolated with RNeasy mini kit (Qiagen) and reverse-transcribed with iScript cDNA synthesis kit (Biorad). Quantitative real-time qRT-PCR was performed using LightCycler 480 SYBR Green Master (Roche) and different sets of primers on LightCycler 480 Real-Time PCR System (Roche). Samples were normalized on GAPDH and analyzed according to the ΔΔCt method. Primer sequences are provided in Supplementary Table 1.

**Single-cell RNA-Seq analysis**. Publicly available single-cell data were used to infer a transcriptome signature of ITGB8-expressing T cells and Treg cells across

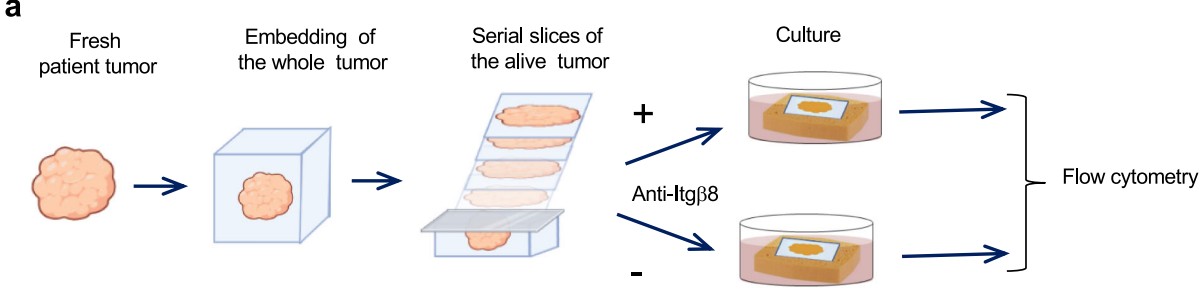

**a**

Fresh patient tumor → Embedding of the whole tumor → Serial slices of the alive tumor → Culture → Flow cytometry

+ Anti-Itgβ8 −

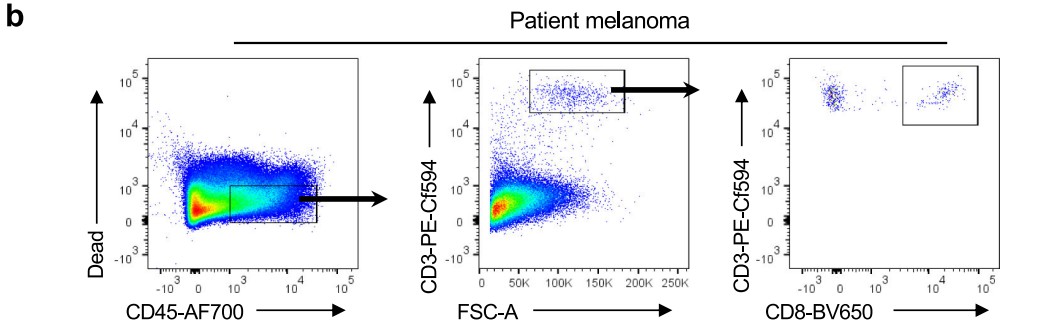

**b**  Patient melanoma

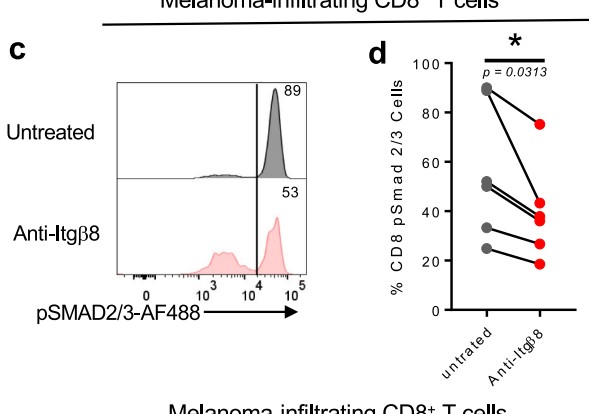

Melanoma-infiltrating CD8⁺ T cells

**c** Untreated 89 / Anti-Itgβ8 53 / pSMAD2/3-AF488

**d** % CD8 pSmad 2/3 Cells; p = 0.0313; * ; untreated, Anti-Itgβ8

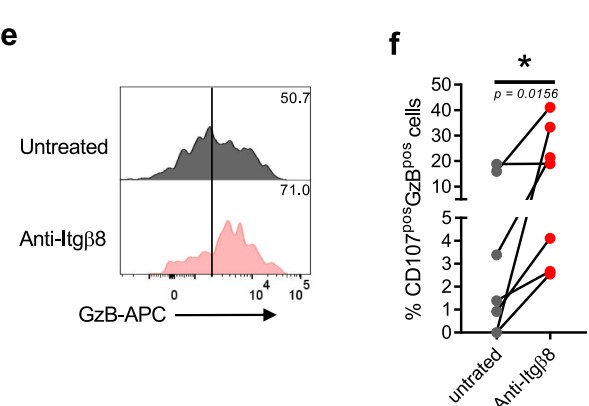

Melanoma-infiltrating CD8⁺ T cells

**e** Untreated 50.7 / Anti-Itgβ8 71.0 / GzB-APC

**f** % CD107ᵖᵒˢGzBᵖᵒˢ cells; p = 0.0156; * ; untrated, Anti-Itgβ8

different tumor types. scRNAseq counts were downloaded from the GEO repository for melanoma (SKCM: GSE115978), colorectal cancer (CCA: GSE108989), liver cancer (HCC: GSE98638), and non-small cell lung cancer (NSCLC: GSE99254). All single-cell data were analyzed with the "Seurat" package v.3.1.0[40]. After creating a seurat object, the sctransform wrapper was used for normalization, scaling, and identification of variable features within each dataset. UMAP projections were used for the visualization of expression and co-expression patterns. The "FindMarkers" function was used with standard settings to identify genes differentially expressed in *ITGB8*ᵖᵒˢ Tregs (CCA, HCC, and NSCLC). For each set of

*ITGB8* single-cell markers, the singscore gene signature scoring method was used to score *ITGB8*ᵖᵒˢ Treg activity (Foroutan et al.[41]). The singscore method uses rank-based statistics to analyze the sample's gene-expression profile and scores the expression activities of gene sets at a single-sample level.

**TCGA data exploitation**. Melanoma (SKCM), colon cancer (COAD), liver cancer (LIHC), and lung cancer (LUAD) gene expression and clinical data were downloaded from the The Cancer Genome Atlas (TCGA) repository [https://

**Fig. 7 Targeting Itgβ8 in melanoma patients impairs TGF-β signaling and increases cytotoxic functions in tumor-infiltrating CD8pos T cells.** Fresh serial sections from the same melanoma for each patient were maintained in culture conditions in the presence or not of neutralizing anti-Itgβ8 antibody as illustrated in (**a**). **b**–**e** Forty-eight hours later, CD8pos T cells infiltrating the tumors were analyzed by flow cytometry as illustrated in (**b**). **c** Representative histograms of the levels of phosphorylated SMAD2/3 (p-SMAD2/3) in tumor CD8pos T cells in the response to anti-Itgβ8 antibody treatment. Percentages of cells with high levels of phosphorylation of SMAD2/3 are indicated. **d** Graph illustrates the percentage of CD8pos T cells positive for p-SMAD2/3 in response to treatment. Each gray dot illustrates the value for tumor CD8pos T cells in the absence of treatment and the linked red dot that of CD8 T cells from the same tumor after anti-Itgβ8 antibody treatment. **e** Representative histograms of the levels of granzyme B (GzB) in tumor CD8pos T cells in the response to anti-Itgβ8 antibody treatment. Percentages of cells producing high levels of GzB are indicated. **f** Graph illustrates the percentage of CD8pos T cells positive for both GzB and CD107 in response to treatment. Each gray dot illustrates the value for tumor-infiltrating CD8pos T cells in the absence of treatment and the linked red dot that of CD8pos T cells from the same tumor after anti-Itgβ8 treatment. $n = 6$–7 different patients, $*P < 0.05$ was determined by two-tailed Wilcoxon test. **d** $P = 0.0313$ (*), **f** $P = 0.0156$ (*). Source data are provided as a Source Data file.

www.cancer.gov/tcga] using the R packages "TCGAbiolinks" v.2.9.4[42] and "RTCGA.clinical" v.20151101.8.0 (Kosinski[43]), respectively. Briefly, each dataset was queried for Illumina HiSeq RNA-Seq results. Downloaded data were pre-processed, normalized and filtered using "TCGAbiolinks" and edgeR (Robinson et al.[44]), and a Itgb8pos score was calculated for each sample, as described above. Tumor samples were classified into two groups, Low and High, based on the median of the Itgb8pos score. Survival data (i.e., time to last follow-up and overall survival status), was used to fit a Cox proportional hazards regression model using the "survival" package v.2.44-1.1 [https://CRAN.R-project.org/package=survival]. Survival Kaplan–Meier curves were plotted with "survminer" v.0.4.5 [https://CRAN.R-project.org/package=survminer]. Additional validations and score correlations were also performed in the melanoma expression dataset[45].

**Statistical analysis.** Statistical analysis was performed using paired *t* test, unpaired *t* test, Mann–Whitney or Wilcoxon when appropriate. For survival analysis, Mantel Cox proportional hazard model was used. Differences were considered significant when *P* values were <0.05. Correlation scores were obtained using the Spearman test.

**Reporting summary.** Further information on research design is available in the Nature Research Reporting Summary linked to this article.

## Data availability

Published Single Cell RNA-seq data used in this study was obtained from the Gene Expression Omnibus (GEO) repository under accession codes: GSE115978 (https://doi.org/10.1016/j.cell.2018.09.006), GSE108989 (https://doi.org/10.1038/s41586-018-0694-x), GSE98638 (https://doi.org/10.1016/j.cell.2017.05.035), and GSE99254 (https://doi.org/10.1038/s41591-018-0045-3). Gene expression and survival data were obtained from The Cancer Genome Atlas (TCGA) repository (https://www.cancer.gov/tcga, datasets: SKCM, COAD, LIHC, and LUAD), and the National Center for Biotechnology Information database of Genotypes and Phenotypes (https://www.ncbi.nlm.nih.gov/gap/) with accession number phs001919 (https://doi.org/10.1038/s43018-019-0003-0). All codes used in this study were based on the following R packages: Seurat v.3.1.0 (doi: Article file 070921.doc), singscore v1.10.0 (https://doi.org/10.1186/s12859-018-2435-4), TCGAbiolinks v.2.9.4 (https://doi.org/10.1093/nar/gkv1507), RTCGA.clinical v.20151101.8.0, edgeR v.3.31.1 (https://doi.org/10.1093/bioinformatics/btp616), survival v.2.44-1.1 (https://github.com/therneau/survival), and survminer v.0.4.5 (https://rpkgs.datanovia.com/survminer/index.html). Data supporting the findings of this study are provided either in supplementary figures or in the Source data file. Source data are provided with this paper.

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

## Acknowledgements

The expert assistance of V. Bernet, S. Rodriguez, E. Guillemot, and H. Tarayre is acknowledged as well as the advices and analysis from V. Fontanier. We also thank Drs. S. Soudja and V. Dardalhon for critical comments and the Marie lab members for their helpful discussions. We thank Dr. Bartholin for sharing TGFβRI^CA mice and Prof. Ribas for sharing data on melanoma patients. The help of the core facilities of the CRCL: PIC (Microscopy), Flow cytometry, Anican and B. Manship for editing is also acknowledged. This work was supported by grants from LabEx DEVweCAN ANR investissement d'Avenir ANR-10-LABX-61 (J.C.M.), the Helmholtz-DKFZ-Inserm program (J.C.M.), BMS foundation Grant, and Foncer contre le cancer, ARC grant # 2019-1047, the labelisation ligue Nationale contre cancer (J.C.M. EL-2016 and EL-2021). Agence Nationale de la Recherche ANR-13-PDOC-0019 (H.P.) and People Program (Marie Sklodowska-Curie Actions) of the European Union (PIIF-GA-2012-330432) (H.P.). A.L. was supported by a LabEx DEVweCAN grant and Ligue Nationale Contre le Cancer. O.L. was supported by the Ligue Nationale Contre le Cancer and S.T. by a PhD fellowship from the French Ministry of Higher Education. The SIRIC LYriCAN INCa-DGOS-Inserm_12563 is acknowledged.

## Author contributions

A.L., O.L., and J.C.M. planned, conducted experiments, and analyzed the data. H.H.-V. performed bioinformatic analysis. S.T. and H.P. generated the *Itgb8-dt-Tomato* mice. S.L., A.L., and O.L. performed patient-tumor ex vivo cultures and A.S. tumor-cell injections and double-blind tumor measurements. D.S. provided the anti-Itgβ8 antibody, S.D. patient melanoma, and M.A.T. *Itgb8^{fl/fl}* mice. J.C.M. wrote the manuscript. H.P. and M.A.T. performed comments and corrections. J.C.M. supervised the study.

## Competing interests

The authors declare no competing interests.
