## [Peer Review File · Nature Communications]

Regulatory T cells promote cancer immune-escape through integrin $\alpha\beta8$ -mediated TGF- β activationREVIEWER COMMENTS

Reviewer #1 (Remarks to the Author):

In this work, the authors demonstrated integrin beta 8+ Tregs could help tumor escape from immune surveillance and explained the mechanisms by using elegant experiments. They showed that beta8+ Tregs could activate TGF-beta1 produced by the cancer cells, leading to the suppression of the cytotoxic function of CD8+ T cells (CD107 and GzB expression) in the tumors, consequently lost of the efficient control of the tumor growth. Most importantly, the authors applied those findings into different cancers and tested the effects of blocking integrin beta 8 on fresh patient tumors.

Major points:

1. The authors suggested integrin beta8+ Tregs could activate TGF-beta1 to the suppression of the cytotoxic function of CD8+ T cells in the tumors environment based on the production of granzyme B cytotoxic granules (GzB) in association with the surface expression of CD107. It would be helpful to block GzB and CD107 on CD8 T cells in Foxp3ΔItgb8 and compare the effects with Foxp3Ctrl mice.
2. NK cells also express CD107 and GzB. The authors need to test NK cells in the Foxp3ΔItgb8 mice.

Minor points:

1. In figure 1, the authors concluded that "among host cells composing the TME, Itgb8pos cells were mainly (85-95%) CD45pos hematopoietic cells (Figure 1A-B)". It's hard to get this information from figure 1A. It would be great if the authors could also calculate the percent of beta8+CD45- cells.
2. It's interesting that integrin beta8+ Tregs plus TGFbCA T cells showed reduced CD107 and GzB in figure 4D, the authors could discuss it.

Reviewer #2 (Remarks to the Author):

This an interesting and timely manuscript that addresses the role of integrin avb8 in activation of TGF-b in the tumor microenvironment. The authors make the following claims:

1. That Tregs represent the principal expressers of avb8 in the tumor microenvironment
2. That Itgb8+ Tregs activate TGF-b to suppress cytotoxic CD8 T cell responses in the tumor
3. That latent TGF-b is provided largely by tumor cells
4. That expression of Itgb8 by Tregs in patient tumors correlates with survival.

Overall the experiments and data presented support these claims, and the experiments and tools used are appropriate and well presented. The study raises some additional interesting questions for future studies. For example, where does the Treg:CD8 interaction occur? Does this require direct Treg: CD8 T cell interaction and/or a DC or other professional APC, or does this occur on the tumor cell?

Comments, questions and suggestions are outlined below:

1. The model proposed by the authors suggests that Tregs directly suppress CD8 T cells by activating TGF-b which then signals to the CD8 T cell. A shortcoming of the current study is the lack of a demonstration of direct Itgb8-dependent Treg suppression of CD8 T cells in vitro. It would improve the paper to show this, although it is not essential for publication.
2. In figure 1, the authors show FACS plots of Itgb8 expression in CD45+ T cells and suggest that other non-immune tumor cells do not express Itgb8. However, it is not clear what other tumor environment cells are included in the extraction and FACS analysis. Can the authors include a plot of all cells (eg FSC/ SSC plot) to show which cells are included in this analysis?

3. Figs 1B, D, F use pie charts – these are not helpful here as they do not provide any indication of variability between tumors. Can these be shown as plots of % CD45+ etc with individual points per tumor.
4. Fig 4 A: could the authors include an unstained or isotype control for the antibody staining. Also, the % of SMAD3+ cells in the tdLN are almost 100%. This seems high – is there a control that can be used here to confirm this? Is this true for all T cells in all LNs, or just those that drain the tumor?
5. Fig 4: D,E. Based on Figure E, the ‘representative’ FACS plots seem to show the samples with the lowest % of CD107 cells. As the % of CD107 cells is quite variable in these experiments and approaches the levels seen in the Treg dItgb8/ TGFbRI wt CD8 transfers, the authors should include all 4 mouse groups in the plots in Fig 4E. The lack of labels of the samples used in 4E and F also make these figures a little hard to understand at first glance.
6. Fig 4F: Can the authors include data for tumor growth in the equivalent control experiments (ie transfer of dItgb8 Tregs with wt TGFbRI T cells). These are needed to confirm that the TGFbRI T cells reduce tumor burden when not suppressed by TGF-b in this T cell transfer model.
7. In some cases the numbers of mice/ independent experiments are a little low – overall the effects and results look convincing but there is considerable variability and uncertainty over some results – for example Fig 4C-F are from only 3-4 mice per group and 2 independent repeats and show considerable variation with a SD of around 30%. These results are critical to the authors conclusions. Ideally experiments would be performed at least 3 times, and for experiments with low numbers of mice, combined data from multiple experiments, or data from all repeats should be shown.
8. In Fig 5 D, the levels of CD107 and % of positive T cells are much lower than in previous experiments. Is this just due to variability in CD107 staining/ gating, or is there a fundamental difference in T cell activation in this model?

Minor points:

1. Error in Fig 2 ‘mesurable’ should be ‘measurable’
2. The figure legend in Fig 4 refers to C, D and E when it should be D, E and F.

Reviewer #3 (Remarks to the Author):

In this manuscript, the authors identified a population of Itgβ8+Treg cells as a key player in the tumor microenvironment to activate TGF-β produced by the cancer cells, which contribute to the suppression of CD8 T cell-mediated tumor cytotoxicity. The human relevance of this finding was confirmed by showing increased CD8 response following treatment of neutralizing anti-Itgβ8 antibody in fresh serial sections of melanoma patient samples, as well as by the negative correlation of high Itgβ8 score extracted from single cell RNAseq data with patient survival in the TCGA melanoma database. The main part of this study focused on the B16 transplantation model of melanoma, thus the generality of the findings might also be restricted considering the inherited drawbacks of transplantation tumor models in studying immune responses. In addition, additional experiments need be done to further test their hypothesis.

Major questions:

In addressing the hypothesis that the TGF-β activated by Itgβ8+Treg cells suppresses CD8+ T cell cytotoxic functions directly in the TME, the authors showed that co-transfer of TregΔItgb8 cells increased the cytotoxic features of WT CD8+ T cells, compared to co-transfer of WT Treg cells, and this phenotype is abolished if the CD8+ T cells have constantly activated TGF-β signaling pathway. However, this piece of data itself could not support the claim that Itgβ8+Treg cells suppresses CD8+ T cell cytotoxic functions directly through TGF-β, as the constantly activated TGF-β signaling pathway could have a dominant effect on suppressing CTLs. The authors should phenotype markers downstream of TGF-β signaling, such as CD103, to investigate whether TGF-β signaling is indeed altered in Foxp3ΔItgb8 mice in CTLs. The authors could also perform direct loss-of-function experiments with TGF-β receptor-deficient CD8 T cells and see whether lacking of TGF-β signaling could suppress tumor growth.

Similar CD8⁺ T cell profiling, e.g. CD103 expression, should also be done in experiments with TGF β -KO tumors. In addition, the phenotype of tumor-infiltrating CD8 T cells in Foxp3 Δ Itgb8 mice injected with TGF β -KO tumors should be analyzed, and the impact on tumor growth needs to be analyzed compared to other groups.

Minor questions:

What is the authors' explanation on why LAP is increased in the TME of Foxp3 Δ Itgb8 mice? Does this suggest increased TGF- β secretion in Foxp3 Δ Itgb8 mice in the TME?

Figure 4F has no figure legend, nor was it mentioned in the main text.

In Figure 3F, to compare the CD107/GzB profile of CD8 T cells in the tumor draining lymph node and draw the conclusion that Itg β 8⁺Treg cells selectively represses the cytotoxic functions of CD8 T cells in the TME is, in my opinion, not a fair comparison, as there are few cytotoxic CD8 T cells in the lymph nodes. If the authors want to make conclusions on how Itg β 8⁺Treg cells affect CD8 T cell activation and response to TGF- β signaling, they could look into markers such as CD44, CD69, CD62L and CD103.

The author mentioned in discussion that it is likely that Tregs control TGF- β signaling in CD8 T cells in close vicinity. In the tumor of Itgb8-td-Tomato/FOXP3-IRES-GFP double reporter mice, could any colocalization between Itg β 8⁺Treg cells and CD8⁺ T cell be observed?

Reviewer #1:

In this work, the authors demonstrated integrin beta 8+ Tregs could help tumor escape from immune surveillance and explained the mechanisms by using elegant experiments. They showed that beta8+ Tregs could activate TGF-beta1 produced by the cancer cells, leading to the suppression of the cytotoxic function of CD8+ T cells (CD107 and GzB expression) in the tumors, consequently lost of the efficient control of the tumor growth. Most importantly, the authors applied those findings into different cancers and tested the effects of blocking integrin beta 8 on fresh patient tumors.

We thank the Reviewer #1 for his/her time and the useful comments to improve the original version of our work. We also appreciate that Reviewer #1 underlined that elegance of our experiments and the relevance of our data in mice to different human cancers.

Major points:

1. The authors suggested integrin beta8+ Tregs could activate TGF-beta1 to the suppression of the cytotoxic function of CD8+ T cells in the tumors environment based on the production of granzyme B cytotoxic granules (GzB) in association with the surface expression of CD107. It would be helpful to block GzB and CD107 on CD8 T cells in Foxp3ΔItgb8 and compare the effects with Foxp3Ctrl mice.

As mentioned by the Reviewer #1, in Foxp3ΔItgb8 mice the impairment of the tumor growth (Figure 2 A-F) was associated with both exacerbated cytotoxic functions (GzB and CD107) of CD8 T cells in the TME (figure 3C) and massive apoptotic death in the tumors (Figure 3D-E). We appreciated the Reviewer #1 suggestion to block GzB and CD107 on CD8 T cells in Foxp3ΔItgb8 mice. However, according to our knowledge there is no drug to block both GzB and CD107 expression *in vivo*. The only alternative will be to generate a double conditional knock out both genes (GzB and CD107) and perform transfer of Tregs from Foxp3ΔItgb8 mice in the generated animals. Such an experimental approach will take at least 1.5 year removing the timely feature of our work.

In order to confirm the importance of CD8 T cells in the control of tumor growth in Foxp3ΔItgb8 mice, we selectively depleted the CD8 T cells in these animals. We observed that in the absence of CD8 T cells, the control of tumor growth in Foxp3ΔItgb8 mice was totally lost (figure3A). This set of experiments demonstrates the major role of CD8 T cells in the control of tumor growth in Foxp3ΔItgb8 mice. We agree that our data cannot firmly establish that it is exclusively through their cytotoxic program (GzB and CD107) that CD8 T cells control the tumor growth. In line with this, we did not write such a sharp conclusion. Moreover, the supl figure 2 of the original manuscript showed that IFN-γ production was also increased in CD8 T cells from the TEM of Foxp3ΔItgb8 mice compared to those from control mice.

Thus, we agree with the Reviewer #1 that the blocking both Gzb production and degranulation would be helpful. However, we are unable to address this question *in vivo* without long term breeding experiments. We believe that our experiments on Gzb/CD107 staining and on CD8 T cell depletion bring strong evidence for a role of CD8 T cell cytotoxic function in the control of the tumor growth in Foxp3ΔItgb8 mice but do not firmly establish that it is only mechanisms used by CD8 T cells control tumor growth in Foxp3ΔItgb8 mice. This idea is reinforced by the increase of IFN-γ production in CD8 T cells from Foxp3ΔItgb8 mice. To avoid the future readers to draw the conclusion that the tumor growth in Foxp3ΔItgb8 mice involves only GzB and CD107, we discussed this point page 8 of the revised version.

2. NK cells also express CD107 and GzB. The authors need to test NK cells in the *Foxp3ΔItgb8* mice.

We totally agree with the Reviewer #1 that, like CD8 T cells, NK cells express both GzB and CD107. As illustrated in the sup figure 1 of the original manuscript, our analysis on NK cells failed to show any difference in the infiltration of the tumors between *Foxp3ΔItgb8* mice and control mice. Moreover, the simple deprivation of CD8 T cells was sufficient to fully reverse the effect observed on tumor growth in *Foxp3ΔItgb8* mice (figure 3A and B of the original manuscript), demonstrating that CD8 T cells are the main effector cells of the anti-tumor effects observed in *Foxp3ΔItgb8* mice. In order to avoid any confusion on the role of NK cells, we underlined these important points page 6 of the revised manuscript.

Minor points:

1. In figure 1, the authors concluded that “among host cells composing the TME, *Itgb8*pos cells were mainly (85-95%) *CD45*pos hematopoietic cells (Figure 1A-B)”. It’s hard to get this information from figure 1A. It would be great if the authors could also calculate the percent of *beta8+CD45-* cells.

We appreciate the Reviewer #1 suggestion which highly improves the reading of the figure 1A. The percentage of *Itgb8*dt Tomato^{pos} cells among the *CD45*^{neg} cells has been added on figure 1A and histogram 1B now shows the distribution of *Itgβ8* in the *CD45*^{neg} and *CD45*^{pos} compartment in the TME.

2. It's interesting that integrin *beta8*+ Tregs plus *TGFβCA* T cells showed reduced *CD107* and *GzB* in figure 4D, the authors could discuss it.

We thank the Reviewer #1 for the suggestion to discuss this aspect and the requested discussion has been added page 8. We particularly reminded the importance of TGF-β signaling in control of the anti-tumor cytotoxic program of CD8 T cells by quoting Thomas et al Cancer Cell 2002.

Reviewer #2:

This an interesting and timely manuscript that addresses the role of integrin avb8 in activation of TGF-b in the tumor microenvironment. The authors make the following claims:

- 1. That Tregs represent the principal expressers of avb8 in the tumor microenvironment***
- 2. That Itgb8+ Tregs activate TGF-b to suppress cytotoxic CD8 T cell responses in the tumor***
- 3. That latent TGF-b is provided largely by tumor cells***
- 4. That expression of Itgb8 by Tregs in patient tumors correlates with survival.***

Overall the experiments and data presented support these claims, and the experiments and tools used are appropriate and well presented. The study raises some additional interesting questions for future studies. For example, where does the Treg:CD8 interaction occur? Does this require direct Treg: CD8 T cell interaction and/or a DC or other professional APC, or does this occur on the tumor cell?

Comments, questions and suggestions are outlined below:

We appreciated that the Reviewer #2 comments underlying that our work is timely as well as the positive comments on the quality of our data and of the experiment design. We also share with Reviewer #2 the vision that this work paves the path to future interesting studies that will be addressed by our lab in further publications. We thank the Reviewer#2 for her/his time and the suggestions he/she made that largely improved the first version of our manuscript.

1. The model proposed by the authors suggests that Tregs directly suppress CD8 T cells by activating TGF-b which then signals to the CD8 T cell. A shortcoming of the current study is the lack of a demonstration of direct Itgb8-dependent Treg suppression of CD8 T cells in vitro. It would improve the paper to show this, although it is not essential for publication.

Like Reviewer #2, we first thought that *in vitro* data could improve our manuscript. However, *in vitro* cytotoxic tests require culture medium with fetal calf serum (FCS) (5-10%) which constitutes a large source of TGF- β . Moreover, the heat, used for the serum inactivation, is known to increase the release of the active form of this cytokine (Shi et al Nature 2011 [10.1038/nature10152](https://doi.org/10.1038/nature10152)). Finally, T effector cells from Foxp3 Δ Itgb8 mice respond to TGF-b and culture in enriched FCS medium leads to the Smad2/3 branch activation. Hence, culture condition associated with cytotoxic tests are not optimal in this cellular system. Taking in consideration the strong side effect of *in vitro* approaches on the effector cells, we agree with the Reviewer #2 that the presence of this *in vitro* experiment in the manuscript is not essential for publication regarding that our work is exclusively focused on *in vivo* approaches including the maintenance of the TME integrity.

2. In figure 1, the authors show FACS plots of Itgb8 expression in CD45+ T cells and suggest that other non-immune tumor cells do not express Itgb8. However, it is not clear what other tumor environment cells are included in the extraction and FACs analysis. Can the authors include a plot of all cells (eg FSC/SSC plot) to show which cells are included in this analysis?

As illustrated in figure 1A-B and mentioned in the text, the CD45^{neg} cells represent a very small fraction of the Itg β 8^{pos} cells in itgb8 dt-Tomato reporter mice. We are happy to provide to the Reviewer #2 the requested FSC/SSC counterplots on this rare population for both breast cancer (0.030%) and melanoma (0.096%).

Figure 1 R1: Size and morphology of CD45^{neg} Itgb8^{pos} cells in the TME

Itgb8-td-Tomato reporter mice were injected with melanoma cells (B16) or breast cancer cells (E0771) in the dermis or in the mammary gland respectively. 18 days later tumors were analyzed by flow cytometry. Gating strategy is illustrated and representative FSC/SSC counterplot are illustrated for CD45^{neg} cells expressing or not the Itgb8. As illustrated not obvious variation is observed between the two groups melanoma based. Please note that for the melanoma condition, the difference is likely due to the view cells since CD45^{neg} Itgb8^{pos} cells represent 0.03%.

3. Figs 1B, D, F use pie charts – these are not helpful here as they do not provide any indication of variability between tumors. Can these be shown as plots of % CD45+ etc with individual points per tumor.

We appreciate Reviewer #2 suggestion. In figure 1B, D, F of the revised manuscript, pie charts have been replaced by histogram bars showing all individual points per tumor.

4. Fig 4 A: could the authors include an unstained or isotype control for the antibody staining. Also, the % of SMAD3+ cells in the tdLN are almost 100%. This seems high – is there a control that can be used here to confirm this? Is this true for all T cells in all LNs, or just those that drain the tumor?

We apologize for not showing the FMO staining in the figure 4A. The figure has been completed with this internal control.

In T cells, the TGF- β signaling is highly activated explaining why 100% of cells are positive for P-SMAD2/3. This observation has been made by others (please see figure 2 of Donkor et al Immunity 2011 [10.1016/j.immuni.2011.04.019](https://doi.org/10.1016/j.immuni.2011.04.019)). Of note the SMAD2/3 phosphorylation has been reported as even exacerbated in the LNs by the Ming O. Li lab in Donkor et al. Immunity 2011.

5. Fig 4: D,E. Based on Figure E, the ‘representative’ FACS plots seem to show the samples with the lowest % of CD107 cells. As the % of CD107 cells is quite variable in these experiments and approaches the levels seen in the Treg dItgb8/ TGFBR1 wt CD8 transfers, the authors should include all 4 mouse groups in the plots in Fig 4E. The lack of labels of the samples used in 4E and F also make these figures a little hard to understand at first glance.

We agree with Reviewer #2 comment on CD107. Being associated with the degranulation the levels of CD107 at the CD8 T cell surface is quite labile and variable, but reflecting functional degranulation. As Reviewer #2 noticed, in the figure 4D the lower levels

for CD107 were chosen for illustration of all conditions to be constant. To remove any concerns due to the variable expression of CD107, we performed another set of experiments. Reformatted figure 4D with points close to the average value for each conditions and added more points in the figure 4E of the revised manuscript as requested by the Reviewer#2.

We apologize for the absence of clear labelling of 4E and 4F. This point has been fixed in the revised version of the manuscript of the figure 4 legend.

6. Fig 4F: Can the authors include data for tumor growth in the equivalent control experiments (ie transfer of dITtgb8 Tregs with wt TGFbRI T cells). These are needed to confirm that the TGFbRI T cells reduce tumor burden when not suppressed by TGF-b in this T cell transfer model.

We appreciate the Reviewer #2 suggestion to improve our work and the requested data have been added in figure 4F.

7. In some cases the numbers of mice/ independent experiments are a little low – overall the effects and results look convincing but there is considerable variability and uncertainty over some results – for example Fig 4C-F are from only 3-4 mice per group and 2 independent repeats and show considerable variation with a SD of around 30%. These results are critical to the authors conclusions. Ideally experiments would be performed at least 3 times, and for experiments with low numbers of mice, combined data from multiple experiments, or data from all repeats should be shown.

As mentioned at point #5 another set of experiment has been performed and figures completed. In agreement with our animal care committee, the repetition of experiments with multiple cell injections (T cells, and tumor cell lines) should be limited to three times.

8. In Fig 5 D, the levels of CD107 and % of positive T cells are much lower than in previous experiments. Is this just due to variability in CD107 staining/ gating, or is there a fundamental difference in T cell activation in this model?

Compare to other figures, in Fig 5D the tumor cell lines we used are different. B16F10 Δ TGFb1 cells and their empty vector control were implanted in fig 5D. We agree with the Reviewer #2 that the levels of CD107 at the surface of CD8 T cells were lower in Foxp3 Δ Itgb8 mice with these mutated cell lines than thus observed with parental B16F10 in Foxp3 Δ Itgb8 mice. This can be due to cell line variability of immunogenicity after the introduction of the molecular construct. Indeed, the expression of GzB was much higher in fig 5D. Given that CD107 expression at the cell surface is extremely labile, and disappears once the cells have finished to degranulated, the weak surface expression of CD107 can be explained by the exacerbated activation of CD8 T cell cytotoxic function in these TME. Though the description of TGFb1^{CTRL} was depicted in the methods, to avoid reader confusion with parental B16F10 we added this comment in the figure legend of Fig5.

Minor points:

- 1. Error in Fig 2 ‘mesurable’ should be ‘measurable’**
- 2. The figure legend in Fig 4 refers to C, D and E when it should be D, E and F.**

We thank the Reviewer #2 for mentioning the typo mistakes. These latter have been corrected in the revised version of the manuscript.

Reviewer #3

In this manuscript, the authors identified a population of Itgβ8+Treg cells as a key player in the tumor microenvironment to activate TGF-β produced by the cancer cells, which contribute to the suppression of CD8 T cell-mediated tumor cytotoxicity. The human relevance of this finding was confirmed by showing increased CD8 response following treatment of neutralizing anti-Itgβ8 antibody in fresh serial sections of melanoma patient samples, as well as by the negative correlation of high Itgβ8 score extracted from single cell RNAseq data with patient survival in the TCGA melanoma database. The main part of this study focused on the B16 transplantation model of melanoma, thus the generality of the findings might also be restricted considering the inherited drawbacks of transplantation tumor models in studying immune responses.

We thank Reviewer #3 for his/her time and underling the relevance of our data to the human pathology using analysis on both melanoma and breast cancer in mice (figure 1 and 2) and in humans (sup figure 6).

Major questions:

1: In addressing the hypothesis that the TGF-β activated by Itgβ8+Treg cells suppresses CD8+ T cell cytotoxic functions directly in the TME, the authors showed that co-transfer of TregΔItgb8 cells increased the cytotoxic features of WT CD8+ T cells, compared to co-transfer of WT Treg cells, and this phenotype is abolished if the CD8+ T cells have constantly activated TGF-β signaling pathway. However, this piece of data itself could not support the claim that Itgβ8+Treg cells suppresses CD8+ T cell cytotoxic functions directly through TGF-β, as the constantly activated TGF-β signaling pathway could have a dominant effect on suppressing CTLs. The authors should phenotype markers downstream of TGF-β signaling, such as CD103, to investigate whether TGF-β signaling is indeed altered in Foxp3ΔItgb8 mice in CTLs. The authors could also perform direct loss-of-function experiments with TGF-β receptor-deficient CD8 T cells and see whether lacking of TGF-β signaling could suppress tumor growth.

We agree with Reviewer #3 that the constantly activated TGF-β (RCA) signaling pathway could a dominant effect on suppressing CTLs. Hence, if the action mode of Itgβ8^{neg} Treg involves other mechanisms than suppressing the CTL activity, Treg ΔItgβ8 should sustain their suppressive effect on tumor growth with RCA effector T cells. Data exposed in Figure 4F of the original manuscript rule out this hypothesis.

We appreciate the suggestion of using CD103 surface expression as a maker of TGF-β signaling activation. Indeed CD103 expression has been reported as influenced by TGF-β signaling. However, to our knowledge the expression of CD103 in the TME of effector CD8 T cells is quiet late and particularly restricted to long term resident cells. Here tumors were analyzed between 15-18 days after cell injections. Moreover, the control of CD103 on CD8 T cells by TGF-b is largely balanced by the TGF-b signaling branches down-stream of the phosphorylation of SMAD2/3. One example is the works from Kaech lab and Cauley lab in 2015 revealing that SMAD4 represses CD103 on CD8 T cells DOI: [10.4049/jimmunol.1402369](https://doi.org/10.4049/jimmunol.1402369)

In contrast, it is undeniable that the phosphorylation of SMAD2/3 reflects directly the levels of activation of TGF- β signaling. For these reasons, we analyzed the early branch of TGF- β signaling and not the expression of molecules that could be placed under TGF- β control.

We are confused by Reviewer #3 suggestion to use TGF- β Receptor-deficient mice and see whether lacking of TGF- β signaling could suppress tumor growth. Indeed, the importance of TGF- β R in T effector cells for tumor growth progression has been extensively documented since 2001 Gorelik et al Nature Medicine doi: 10.1038/nm1001-1118

2/ Similar CD8+ T cell profiling, e.g. CD103 expression, should also be done in experiments with TGF β -KO tumors.

For the reasons exposed in our answer to point #1, we believe that P-SMAD2/3 analysis is more informative on TGF- β signaling activation than the level of potential target genes such as CD103.

3/ In addition, the phenotype of tumor-infiltrating CD8 T cells in Foxp3 Δ Itgb8 mice injected with TGF β -KO tumors should be analyzed, and the impact on tumor growth needs be analyzed compared to other groups.

We are confused by the suggestion of Reviewer #3 to analyze the phenotype of tumor-infiltrating CD8 T cells in Foxp3 Δ Itgb8 mice injected with TGF β 1-KO tumors since this analysis was illustrated by figure 5D and E of the original version of the manuscript. In addition, Sup figure 3B illustrates the impact of the deprivation of TGF- β 1 in tumor growth. The data are exposed in comparison with tumor clone control transfected with empty vector. We are confused by the potential control (other groups) that could be missed to demonstrate the TGF- β 1 expression in tumor cells is important to control tumor growth.

We believe that all this miss-understandings were due a lack of explanation in the original versions and provided more information in the revised version page 9

Minor questions:

What is the authors' explanation on why LAP is increased in the TME of Foxp3 Δ Itgb8 mice? Does this suggest increased TGF- β secretion in Foxp3 Δ Itgb8 mice in the TME?

We are sorry if our explanation page 9 were not clear. LAP representing the inactive form and being stored on the extracellular matrix in the TME, higher levels of LAP in Foxp3 Δ Itgb8 mice suggest that TGF- β is not activated. To rule out any effect on TGF- β 1 production, we completed our data on figure 5 with analysis showing no difference of *Tgf- β 1* expression in tumors from Foxp3 Δ Itgb8 mice and control mice. We appreciate reviewer #3 suggestion

Figure 4F has no figure legend, nor was it mentioned in the main text.

We apology for the lack of information regarding figure 4F. In agreement with other reviewers the figure 4F have been modified in the revised version

In Figure 3F, to compare the CD107/GzB profile of CD8 T cells in the tumor draining lymph node and draw the conclusion that Itg β 8+Treg cells selectively represses the cytotoxic functions of CD8 T cells in the TME is, in my opinion, not a fair comparison, as there are few cytotoxic CD8 T cells in the lymph nodes. If the authors want to make conclusions on how Itg β 8+Treg cells affect CD8 T cell activation and response to TGF- β signaling, they

could look into markers such as CD44, CD69, CD62L and CD103.

We appreciate reviewer #3 suggestion. We agree with the reviewer #3 that cytotoxic cells are less abundant in the draining LN than in the tumor in wild type mice. However, this information was unknown for the Foxp3 Δ Itgb8 mice. Indeed several other mouse models leading to an exacerbated cytotoxic function in the tdLN were depicted (Poggio et al Cell 2019 doi:10.1016/j.cell.2019.02.016.) and photoconversion approaches revealed important recirculation of effector cells from the TME to the tdLN (Trocellan et al PNAS 2015 [/doi.org/10.1073/pnas.1618446114](https://doi.org/10.1073/pnas.1618446114)).

The CD69 and CD62L are early activation markers and thus their analysis in the LN after several weeks post cells implantation is complicated to interpret particularly knowing recirculation of the cells.

The response to TGF- β signaling both in the tumor and in the tdLN was analyzed in figure 4A and B of the original version by monitoring P-SMAD2/3.

The author mentioned in discussion that it is likely that Tregs control TGF- β signaling in CD8 T cells in close vicinity. In the tumor of Itgb8-td-Tomato/FOXP3-IRES-GFP double reporter mice, could any colocalization between Itgb8+Treg cells and CD8+ T cell be observed?

We appreciate the question of the localization of Treg Itgb8 and CD8 T cells. Tumor slides analysis can show some Tregs in the close vicinity to CD8 T cells as reported by others including human tumors (Curiel et al Nature Medicine 2004). Regarding that these “contacts” are rare events on tissue section (Curiel et al Nature Medicine 2004, and confirmed in our experimental system), no conclusions can be drawn. In order to address this question, we are currently developing bi-photonic microscopy on alive tumors and data will be presented in an another publication depicting molecular mechanisms.

REVIEWERS' COMMENTS

Reviewer #1 (Remarks to the Author):

The authors have addressed my previous comments.

Reviewer #2 (Remarks to the Author):

The authors have addressed all of my concerns

Reviewer #3 (Remarks to the Author):

The authors have clarified most of my concerns. Regarding the conclusion from the authors. In the study, the observation was mainly made from transplantation tumor models, specifically, B16 melanoma cells. The authors should be cautious about how general the conclusions from this model can be. One caveat about using this model is that the tumor grows fast, and the tumor architecture, microenvironment, as well as immune cell composition and their functional importance, could not fully recapitulate the nature of tumor development. For example, it has been shown that CD103+ CD8+ tissue-resident memory type of CTLs, which require TGF- β signaling for their differentiation, mediate important antitumor function and correlates with improved clinical response in several cancer types (1-2). Such a perspective of TGF- β 's contribution to antitumor immunity through regulating Trm is omitted in this fast-harvesting transplantation tumor model. Another recent study also points out that tumor-infiltrating CD103+ CD8+ T cells themselves could upregulate β 8 and produce active TGF- β , sustaining their CD103 expression, granting them enhanced cytotoxicity (3). Thus, the efficacy of targeting β 8 with anti- β 8 antibody might require more considerations.

Quantification of LAP to Fig 5D needs be included.

Ref

1. Boutet, M., Gauthier, L., Leclerc, M., Gros, G., Montpreville, V. de, Théret, N., Donnadieu, E., & Mami-Chouaib, F. (2016). TGF β Signaling Intersects with CD103 Integrin Signaling to Promote T-Lymphocyte Accumulation and Antitumor Activity in the Lung Tumor Microenvironment. *Cancer Research* , 76 (7), 1757–1769. <https://doi.org/10.1158/0008-5472.can-15-1545>
2. Komdeur, F. L., Wouters, M. C., Workel, H. H., Tijans, A. M., Terwindt, A. L., Brunekreeft, K. L., ... & de Bruyn, M. (2016). CD103+ intraepithelial T cells in high-grade serous ovarian cancer are phenotypically diverse TCR $\alpha\beta$ + CD8 $\alpha\beta$ + T cells that can be targeted for cancer immunotherapy. *Oncotarget* , 7 (46), 75130. [10.18632/oncotarget.12077](https://doi.org/10.18632/oncotarget.12077)
3. Hamid, M. A., Colin-York, H., Khalid-Alham, N., Browne, M., Cerundolo, L., Chen, J.-L., Yao, X., Rosendo-Machado, S., Waugh, C., Maldonado-Perez, D., Bowes, E., Verrill, C., Cerundolo, V., Conlon, C. P., Fritzsche, M., Peng, Y., & Dong, T. (2020). Self-Maintaining CD103+ Cancer-Specific T Cells Are Highly Energetic with Rapid Cytotoxic and Effector Responses. *Cancer Immunology Research* , 8 (2), 203–216. <https://doi.org/10.1158/2326-6066.cir-19-0554>

Point by point answers to Reviewer #3

The authors have clarified most of my concerns. Regarding the conclusion from the authors. In the study, the observation was mainly made from transplantation tumor models, specifically, B16 melanoma cells. The authors should be cautious about how general the conclusions from this model can be. One caveat about using this model is that the tumor grows fast, and the tumor architecture, microenvironment, as well as immune cell composition and their functional importance, could not fully recapitulate the nature of tumor development. For example, it has been shown that CD103+ CD8+ tissue-resident memory type of CTLs, which require TGF- β signaling for their differentiation, mediate important antitumor function and correlates with improved clinical response in several cancer types (1-2). Such a perspective of TGF- β 's contribution to antitumor immunity through regulating Trm is omitted in this fast-harvesting transplantation tumor model. Another recent study also points out that tumor-infiltrating CD103+ CD8+ T cells themselves could upregulate β 8 and produce active TGF- β , sustaining their CD103 expression, granting them enhanced cytotoxicity (3). Thus, the efficacy of targeting β 8 with anti- β 8 antibody might require more considerations.

We agree with Reviewer #3 comment on the importance of CD8 Trm in long term response and that we were unable to address this point through a slow tumor growth. However, we would like take with extreme caution the fact that Itgb8 could be expressed on Trem based on fig1 E of the article (<https://doi.org/10.1158/2326-6066.cir-19-0554>) quoted by the reviewer #3. In indeed, regarding the absence of antibody allowing FACS staining of Itgb8 positive cells and the absence of negative control in the fig1E of Hamid M et al 2020 Cancer Immunology Research , 8 (2), 203–216, it is hard to be convinced that Trm actually express itgb8. We would like to remind that we recently reported that Itgb8 expression on Tregs contributes to CD8 Trm cell development which requires a bioactive source of TGF-b1 (Fereira et al Nature immunology 2020 doi: 10.1038/s41590-020-0674-9).

We completed the discussion by mentioning this point and underlying that further investigations, using mouse models with slower growth than B16-F10 should address whether, in the context of TME, this Itgb8 dependent function of Treg occurs could contribute to long term anti-tumor protection.

Quantification of LAP to Fig 5D needs be included.

We apology for forgetting to show the quantification of LAP. This quantification is now added in figure 5 of the revised manuscript.